# Estimating causal networks in biosphere-atmosphere interaction with the PCMCI approach

Christopher Krich[1,2], Jakob Runge[3], Diego G. Miralles[2], Mirco Migliavacca[1], Oscar Perez-Priego[1], Tarek El-Madany[1], Arnaud Carrara[4], and Miguel D. Mahecha[1,5]

[1]Max Planck Institute for Biogeochemistry, 07745 Jena,Germany
[2]Laboratory of Hydrology and Water Management, Ghent University, Ghent 9000, Belgium
[3]German Aerospace Center, Institute of Data Science, 07745, Jena, Germany
[4]Fundación Centro de Estudios Ambientales del Mediterráneo (CEAM), 46980 Paterna, Spain
[5]German Centre for Integrative Biodiversity Research (iDiv), Deutscher Platz 5e, 04103 Leipzig, Germany

**Correspondence:** Christopher Krich (ckrich@bgc-jena.mpg.de)

**Abstract.** The dynamics of biochemical processes in terrestrial ecosystems are tightly coupled to local meteorological conditions. Understanding these interactions is an essential prerequisite for predicting e.g. the response of the terrestrial carbon cycle to climate change. However, many empirical studies in this field rely on correlative approaches and only very few studies apply causal discovery methods. Here we explore the potential of a recently proposed causal graph discovery algorithm to reconstruct the causal dependency structure underlying biosphere-atmosphere interactions. Using artificial time series with known dependencies that mimic real-world biosphere-atmosphere interactions we address the influence of non-stationarities, i.e. periodicity and heteroscedasticity, on the estimation of causal networks. We then investigate the interpretability of the method in two case studies. Firstly, we analyse three replicated eddy covariance datasets from a Mediterranean ecosystem, and secondly, we explore global NDVI time series (GIMMS 3g) along with gridded climate data to study large-scale climatic drivers of vegetation greenness. We compare the retrieved causal graphs to simple cross correlation-based approaches to test whether causal graphs are considerably more informative. Overall, the results confirm the capacity of the causal discovery method to extract time-lagged linear dependencies under realistic settings. For example, we find a complete decoupling of the net ecosystem exchange from meteorological variability during summer time in the Mediterranean ecosystem. However, cautious interpretations are needed as the violation of the method's assumptions due to non-stationarities increases the likelihood to detect false links. Overall, estimating directed biosphere-atmosphere networks helps unravelling complex multi-directional process interactions. Other than classical correlative approaches, our findings are constrained to a few meaningful sets of relations which can be powerful insights for the evaluation of terrestrial ecosystem models.

# 1 Introduction

The terrestrial biosphere responds to atmospheric drivers such as radiation intensity, temperature, vapour pressure deficit, and composition of trace gases. On the other hand, the biosphere influences the atmosphere via partitioning incoming net radiation into sensible, latent, and ground heat fluxes as well as via controlling the exchange of trace gases and volatile organic compounds. Over the past decades, many of these processes have been identified and their physical, chemical and biological effects have been investigated (see e.g. Monson and Baldocchi, 2014; McPherson, 2007, for overviews). However, there are still substantial unknowns regarding the exact causal dependencies among the different processes (Baldocchi et al., 2016; Miralles et al., 2018), which leads to large uncertainties when predicting e.g. ecosystem responses to drought conditions von Buttlar et al. (2018); Sippel et al. (2017).

Multiple ecological monitoring systems have been setup to monitor ecosystem dynamics. Networks of eddy covariance towers continuously monitor carbon, water, and energy fluxes in high temporal resolution (Baldocchi, 2014). Satellite remote sensing data complement this picture and can be used in tandem (Papale et al., 2015; Mahecha et al., 2017). They typically only monitor vegetation states at multi-day resolutions and some products offer nearly complete global coverage (Justice et al., 2002; Woodcock et al., 2008). The actual and future satellite missions are leading to rapid development in the field with ever higher spatial, temporal, and spectral measurements (Malenovský et al., 2012; Guanter et al., 2015; Qi and Dubayah, 2016).

The study of biosphere-atmosphere interactions using observations typically relies on correlative approaches, or is based on model-data i.e. requires *a-priori* knowledge. In recent years, a new branch in statistics aiming for causal inference from empirical data has experienced substantial progress. The idea of causal inference emerged already in the early 20th century (Wright, 1921). Later, Granger suggested one of the first applicable formalisms (Granger, 1969); since then, several efforts in ecology and climate science have concentrated on the bivariate form of Granger causality (Elsner, 2006, 2007; Kodra et al., 2011; Attanasio, 2012; Attanasio et al., 2012). From an information theoretic perspective transfer entropy (Schreiber, 2000) evolved as a frequently used measure to infer directionality and amount of information flow (Kumar and Ruddell, 2010; Ruddell et al., 2015; Gerken et al., 2018; Yu et al., 2019). For instance, Ruddell and Kumar (2009) used transfer entropy to estimate networks of information flow. These networks constructed for an agricultural site under drought and non-drought conditions showed substantial differences in connectivity, especially between subsystems comprising variables of land and atmospheric conditions. Those changes in connectivity are attributed to changes in the feedback patterns between the subsystems for drought and normal conditions. The original forms of both Granger causality and transfer entropy are bivariate and converge for the case of vector auto regressive models. While Granger causality is typically limited to linear relationships, transfer entropy captures also non-linear interactions, but requires very large data quantities for the estimation of the probability density function.

Aiming to mitigate some of the limitations of the traditional Granger causality, Detto et al. (2012) used a conditional spectral Granger causality framework that allows to disentangle system inherent periodic couplings from external forcing. The disentanglement is enabled via decomposition into the frequency domain using wavelet theory. This method enabled the finding that soil respiration in a pine and hardwood forested ecosystem in winter is not influenced by canopy assimilation but only by temperature, a result that would not be detectable via lagged correlation or bivariate Granger causality. A time-frequency

representation of Granger causality was presented by Shadaydeh et al. (2019) which allowed to identify anomalous events in marine and ecological time series. Green et al. (2017) used a similar approach as Detto et al. (2012) to investigate biosphere-atmosphere feedback loops. It was found that they can explain up to 30% of variance in radiation and precipitation in certain regions. Recently, Papagiannopoulou et al. (2017a) applied a non-linear multivariate conditional Granger causality frame-work to study climatic drivers of vegetation at the global scale. This approach revealed that water availability dominates plant productivity as 61% of the vegetated surface appeared water limited rather than controlled by radiation or temperature (Papa-giannopoulou et al., 2017b). In the case of transfer entropy, Goodwell and Kumar (2017a, b) developed a redundancy measure which allows to distinguish unique, synergistic and redundant information transfer of a bi or potentially multivariate system to a target variable. This modification enables a stronger multivariate interpretation of process networks constructed with transfer entropy. Changes in connectivity then potentially point to different ecosystem response strategies to disturbances (Goodwell et al., 2018). These examples highlight that unexpected interaction patterns can in principle be identified from data only and may challenge theoretical assumptions. In fact, in the last years the science of causal inference has developed a strong theo-retical foundation and several algorithms have been proposed (Spirtes et al., 2001; Pearl, 2009; Peters et al., 2017; Pearl and Mackenzie, 2018; Runge et al., 2019a). However, only few studies test the suitability of this latest generation of methods to understand ecosystem dynamics (see e.g. Shadaydeh et al., 2018, 2019; Christiansen and Peters, 2018).

Ecological and climate data are often time ordered. This property can be exploited to construct time series graphs (Ebert-Uphoff and Deng, 2012). Recently, Runge et al. (2019b) introduced an algorithm to estimate such graphs, called PCMCI, a combination of the PC algorithm (named after its inventors Peter and Clark, Spirtes and Glymour, 1991) and the Momentary Conditional Independence (MCI) test. PCMCI has been successfully applied to artificial (Runge et al., 2018) and climatological case studies (Runge et al., 2014; Kretschmer et al., 2016). Hence, this method could be potentially of high relevance for learning the causal dependency structure underlying biosphere-atmosphere interactions.

In this study, we explore the potential of PCMCI for disentangling and quantifying interactions and feedbacks between terrestrial biosphere state and fluxes and meteorological variables. The study is structured as follows: In Sect. 2 we motivate and introduce the method from an ecological perspective. We also describe artificial and real world datasets explored in this study. The results in Sect. 3 describes the performance of the method on artificial time series data with known dependencies that mimic some basic properties of observed land surface fluxes such as heteroscedasticity. We then report on the explo-ration of three replicated eddy covariance measurement towers in a Mediterranean ecosystem and explore how the identified interdependencies of carbon and energy fluxes and micro-meteorological observations vary over time. Further, we present the analysis of global satellite data of vegetation greenness to understand the lagged dependency of ecosystems with respect to climatic drivers. Based on these results, Sect.4 discusses the potentials and limitations of PCMCI for other applications in land-atmosphere studies and give recommendations for further methodological developments.

## 2 Method and Data

### 2.1 From bivariate to multivariate measures of causality

Monitoring an ecosystem with continuous observations of net ecosystem exchange (NEE), the underlying gross primary production (GPP) and ecosystem respiration ($R_{textnormaleco}$) together with the relevant drivers i.e. global radiation (Rg), surface air temperature (T), and soil moisture (SM), allows to study the dynamics of the carbon cycle in terrestrial ecosystems. To foster its understanding, a fundamental question is how these variables causally depend on each other. This requires the identification of directional dependencies such as the well known effects of SM $\rightarrow$ GPP and GPP $\rightarrow$ $R_{eco}$ and their differentiation from physically implausible links such as $R_{eco} \rightarrow$ GPP. Graphical causal models (Spirtes et al., 2001) provide a framework to represent and identify causal relations based on conditional independence relations in data streams of this kind. In the case of an ecological monitoring site as described here, we can exploit the temporal information of the observations for identifying a time series graph as a type of graphical model (Runge, 2018a). Formally this can be stated as follows: The variables $X_t^i$ comprise a multivariate stochastic process $\mathbf{X}$ (where i is the variable index, in the example $i \in \{Rg, T, GPP, R_{eco}, SM\}$, and t is the time index). A time series graph $\mathcal{G}$ visualizes how the individual variables $X^i \in \mathbf{X}$ depend on each other at specific time lags $\tau$, i.e. $X_{t-\tau}^i$ with $\tau \in \{1, .., \tau_{max}\}$ (see Runge, 2018a, for definitions). In the following, we refer to a variable $X_{t-\tau}^i$ that is causally affecting a variable $X_t^j$ as 'parent' or 'driver' and the latter as 'receiver' or 'target'. To come to a causal interpretation, it is important to exclude dependencies between two variables that are due to common drivers ($X_{t-\tau_1}^i \leftarrow X_{t-\tau_2}^s \rightarrow X_t^j$) or indirect paths ($X_{t-\tau_2}^i \rightarrow X_{t-\tau_1}^s \rightarrow X_t^j$). For instance, when estimating the effects of GPP on $R_{eco}$ and Rg on $R_{eco}$ using a bivariate measure, one likely obtains implausible results like to strong or even unexpected links because T, respectively as the common driver and mediator (indirect path), is not accounted for. To exclude dependencies due to common drivers or indirect paths, conditional independence tests are used, denoted as $CI(X_{t-\tau}^i, X_t^j | \mathcal{S})$, with some conditioning set $\mathcal{S}$. If any variable (or their combination) in $\mathcal{S}$ explains the dependence between $X_{t-\tau}^i$ and $X_t^j$, then the CI statistic is zero.

Two prominent methods that aim for directional dependencies are Granger causality and transfer entropy (Granger, 1969; Schreiber, 2000). Granger causality is typically estimated as a vector autoregressive model and thus captures only linear links. Transfer entropy, based on information theory, captures also non linear dependencies. It can be shown that for multivariate Gaussians, transfer entropy is equivalent to Granger causality (Barnett et al., 2009). Both can be phrased as testing for conditional independence (Runge et al., 2019a). In their original bivariate form, neither of these two methods accounts for third variables. But both can also be extended to deal with multivariate time series as required here (Runge et al., 2012; Granger, 1969). There are even non-linear and spectral modifications of Granger causality which have been applied to study biosphere atmosphere interactions (Papagiannopoulou et al., 2017a; Detto et al., 2012; Claessen et al., 2019). However, the estimation of multivariate transfer entropy is challenging due to the "curse of dimensionality" (Runge et al., 2012) and also multivariate Granger causality exhibits low link detection power for larger number of variables (higher dimensions) and limited sample size, as is the case in our application (Runge et al., 2019b). The strong decrease in detection power happens when using the whole past $\mathbf{X}_t^- = (\mathbf{X}_{t-1}, \mathbf{X}_{t-2}, ...)$ of $X_t^j$, truncated at a maximum lag $\tau_{max}$, as a conditioning set $\mathcal{S}$. The problem is that this set can contain a high number of conditions which are irrelevant. For example, when assessing the effect of Rg at a specific time

lag $\tau$ on GPP using multivariate Granger causality one would create a vector auto regressive model comprising all variables, i.e. Rg, T, SM, GPP and $R_{eco}$ at each available lag. But $R_{eco}$, dominated by heterotrophic respiration, is not expected to affect gross primary productivity and could be removed to decrease the dimensionality. However, manually selecting conditions is not desirable when the underlying dependence structure is unknown which is why ideally the conditioning set is identified automatically.

PCMCI addresses this issue by reducing the set of conditions $\mathcal{S}$ prior to quantifying the dependence between two variables. The two-step approach utilizes a variant of the PC algorithm (Spirtes and Glymour, 1991) and the momentary conditional independence measure (MCI) (Runge et al., 2019b). More detailed descriptions are given in Sect. 2.4 and 2.5, respectively (full description of PCMCI including proofs and quantitative comparisons with other methods are provided in Runge et al. (2019b)). A schematic of the PCMCI approach is given in Fig. A1. PCMCI belongs to the family of causal graphical models (Spirtes et al., 2001; Pearl, 2009), and follows the assumptions listed in Sect. 2.2. In the limit of infinite time series length, PCMCI converges to the true graph of dependencies, which is why we use the term "causal". As we deal with finite sample length and partially unfulfilled assumptions, spurious links can still appear (beyond the expected false positive rate) and therefore each detected link has to be interpreted with caution.

## 2.2 Assumptions

PCMCI assesses the causal structure of a multivariate dataset or process $\mathbf{X}$ by estimating its time series graph. To draw causal conclusions from observational data, any causal method must adopt a number of assumptions (Pearl, 2009; Spirtes et al., 2001). For the time series case, here we assume time order, the causal Markov condition, faithfulness, causal sufficiency, causal stationarity, and no contemporaneous causal effects (Runge et al., 2018). PCMCI is applied in combination with the ParCorr linear independence test based on partial correlations (cf. Sect. 2.3). This application additionally requires stationarity in mean and variance and linear dependencies. In the following, we briefly discuss these assumptions (further details in Runge, 2018a; Runge et al., 2019b).

The time order within the time series allows to orient directed links which are only pointing forward in time. This accounts for causal information propagating forward in time only, i.e. the cause shall precede the effect. Therefore, a directed causal link $X_{t_1}^i \rightarrow X_{t_2}^j$ can only exist between two nodes $X_{t_1}^i, X_{t_2}^j$ if $t_1 < t_2$. When a contemporaneous link is found, i.e. $t_1 = t_2$, it is considered to be undirected. In ecological language it means that in order to claim that Rg is driving GPP any change in Rg that is affecting GPP must be measured at a time step before the change in GPP occurs. The Causal Markov and faithfulness assumptions relate the underlying physical causal mechanisms to statistical relationships manifest in the data. The Causal Markov condition states that if two processes are not directly connected by some physical mechanism, then they should be statistically independent conditional on their direct drivers, like Rg and $R_{eco}$ conditional on T . The faithfulness assumption concerns the other direction: if two processes are statistically independent, then there cannot be a direct physical mechanism. The causal sufficiency assumption implies that every common cause of two or more variables $X^i \in \mathbf{X}$ is included in $\mathbf{X}$. If this is not the case, detected links may be indirect or due to an unobserved common driver. However, the absence of a link in the detected graph still implies that no direct link is present (as this only requires the assumption of faithfulness). For example, Rg

is expected to influence $R_{eco}$ via T, the indirect path. Though, a link between Rg and $R_{eco}$ might be detected if T is not included in the analysis. However, a missing link between T and $R_{eco}$, might indicate conditions inhibiting respiratory processes, i.e. very cold temperatures with frozen surfaces or very dry conditions with dead vegetation coverage. Causal stationarity refers to the existence of links over time. In a deciduous forest, for example, the ecosystem's $CO_2$ exchange is not causally stationary as the link Rg $\rightarrow$ GPP is given in summer but not in winter. Formally, a process $\mathbf{X}$ with graph $\mathcal{G}$ is called causally stationary over a time index $\mathcal{T}$, if and only if for all links $X_{t-\tau}^i \rightarrow X_t^j$ in the graph the condition $X_{t-\tau}^i \not\perp\!\!\!\perp X_t^j \mid \mathbf{X}_t^- \setminus \{X_{t-\tau}^i\}$ holds for all $t \in \mathcal{T}$.

## 2.3 Independence Test

At the core of PCMCI there are conditional independence tests $CI(X_{t-\tau}^i, X_t^j, \mathcal{S})$ to evaluate whether $X_{t-\tau}^i \perp\!\!\!\perp X_t^j \mid \mathcal{S}$ given a conditioning set $\mathcal{S}$. Within the PCMCI software package Tigramite (Runge, 2018b), several independence tests are implemented. Here, we focus on the linear independence test called ParCorr. The ParCorr conditional independence test is based on partial correlations and a t-test. This assumes the model

$$X^i = \mathcal{S}\boldsymbol{\beta}_{X^i} + \epsilon_{X^i}, \quad X^j = \mathcal{S}\boldsymbol{\beta}_{X^j} + \epsilon_{X^j}, \tag{1}$$

with coefficients $\boldsymbol{\beta}$ and Gaussian noise $\epsilon$. This leads to the residuals

$$r^{X^i} = X^i - \mathcal{S}\hat{\boldsymbol{\beta}}_{X^i}, \quad r^{X^j} = X^j - \mathcal{S}\hat{\boldsymbol{\beta}}_{X^j} \tag{2}$$

with estimated $\hat{\boldsymbol{\beta}}$. ParCorr removes the influence of $\mathcal{S}$ on $X^i$ and $X^j$ via ordinary least squares regression and tests for independence of the residuals using the Pearson correlation with a t-test. The independence test returns a $p$-value and test statistic value $I$, i.e. the correlation coefficient in case of ParCorr. Thus, to identify the effect of GPP on $R_{eco}$ that does account for their common driver $T \in \mathcal{S}$, ParCorr will perform a linear regression of T on both GPP and $R_{eco}$ accounting for time lags. The p-value of the residuals' partial correlation test can be used to assess whether the two variables are dependent.

## 2.4 PC algorithm

To efficiently estimate $CI(X_{t-\tau}^i, X_t^j | \mathcal{S})$ the conditioning set $\mathcal{S}$ should be as small as possible which means that it should only contain relevant conditions, which allow to isolate the unique influence of $X_{t-\tau}^i$ on $X_t^j$. For an estimation of $CI(\text{Rg}_{t-\tau}, \text{GPP}_t | \mathcal{S})$, for example, $\mathcal{S}$ should contain T and SM (at certain lags), as they influence the ability of an ecosystem to perform photosynthesis. Likewise, when estimating $CI(\text{T}_{t-\tau}, \text{GPP}_t | \mathcal{S})$, $\mathcal{S}$ should include Rg and SM for the same reasons. A sufficient set of relevant conditions includes the drivers/parents of the variable $X_t^j$. Consequently, the aim of the PC step is to identify an as small as possible superset of the parents of each variable included in the process. The algorithm uses a variant of the PC algorithm (Spirtes et al., 2001); a comprehensive pseudo-code of this procedure is given in the supplementary materials of Runge et al. (2019b). In the limit of infinite sample size the relevant conditions indeed converge to the true causal parents, practically though, an estimate that contains a few irrelevant conditions, like $R_{eco}$, is sufficient as well.

The PC step starts by initializing the whole past of a process: $\widetilde{\mathcal{P}}(X_t^j) = \mathbf{X}_t^- = \{X_{t-\tau}^i : i = 1, ..., N, \tau = 1, ..., \tau_{max}\}$. Next, by evaluating $CI(X_{t-\tau}^i, X_t^j, \mathcal{S})$, conditions $X_{t-\tau}^i$ are removed from $\widetilde{\mathcal{P}}(X_t^j)$ that are independent of $X_t^j$ conditionally on a subset

$\mathcal{S} \in \widetilde{\mathcal{P}}(X_t^j) \backslash \left\{ X_{t-\tau}^i \right\}$. $\mathcal{S}$ starts as the empty set $\emptyset$ and is iteratively increased. For instance, let $X_{t-\tau}^i$ be $R_{eco}$ (at a specific lag) and $X_t^j$ be GPP. The conditional independence between GPP and $R_{eco}$ will be estimated first by using no conditions. If GPP and $R_{eco}$ appear related, one variable will be included in the conditioning set. If the residuals are still dependent, a second variable is included and so on. When T is part of the conditioning set, the residuals of GPP and $R_{eco}$ might not be dependent anymore and $R_{eco}$ is removed from the estimated set of parents of GPP. The PC algorithm adopted in PCMCI efficiently selects those conditioning sets to limit the number of tests conducted.

Every conditional independence test is evaluated at a significance threshold $\alpha_{pc}$, which is usually set to a liberal value between 0.1 and 0.4. Alternatively, in tigramite one can let $\alpha_{pc}$ unspecified. PCMCI then evaluates the best choice of $\alpha_{pc}$ $\in \{0.1, 0.2, 0.3, 0.4\}$ based on the Akaike information criterion which is further explained in (Runge et al., 2018).

## 2.5 MCI tests

MCI is the actual causal discovery step that ascribes a $p$-value and strength to each possible link. MCI iterates through all pairs $(X_{t-\tau}^i, X_t^j) : i = 1, ..., N$, $\tau = 0, ..., \tau_{max}$ and calculates $CI(X_{t-\tau}^i, X_t^j, \mathcal{S})$ where $\mathcal{S}$ consists of the two (super-)sets of parents $\widetilde{\mathcal{P}}(X_t^j)$ and $\widetilde{\mathcal{P}}(X_{t-\tau}^i)$ obtained in the PC step. $\widetilde{\mathcal{P}}(X_{t-\tau}^i)$ is constructed by shifting the time series of $\widetilde{\mathcal{P}}(X_t^i)$ by $\tau$. In case $X_{t-\tau}^i \in \widetilde{\mathcal{P}}(X_t^j)$, $X_{t-\tau}^i$ has to be removed from $\widetilde{\mathcal{P}}(X_t^j)$. If $\tau = 0$, conditional dependence is estimated for contemporaneous nodes $X_t^j$ and $X_t^i$. Due to missing time order, a dependence would be left undirected. Further, as the parents $\widetilde{\mathcal{P}}(X_t^i)$ and $\widetilde{\mathcal{P}}(X_t^j)$ used in each conditional dependence test are defined to lie in the past of $X_t^i$ and $X_t^j$, links, both contemporaneous and lagged, can be spurious due to contemporaneous common drivers or contemporaneous indirect paths. The absence of a link, though, means that a physical (contemporaneous) link is unlikely (assuming faithfulness, cf. Runge et al. (2018)). For simplicity, the previously given examples were omiting the time lag. Thus if $R_{eco}$ responds instantaneously (considering the sampling temporal resolution) to changes in T but T responds with a time lag to Rg, both variables will likely appear contemporaneously coupled to Rg.

The link strength in the PCMCI framework can be given by the effect size of the conditional independence test statistic measure $CI$ used in combination with MCI. In case of ParCorr, the effect size is given by the partial correlation value, which is between -1 and 1. Assuming a linear Gaussian model the partial correlation value is shown to directly depend on the receiver's and driver's variance as well as the coupling coefficient (Runge et al., 2019a):

$$\rho_{X_{t-\tau}^i \to X_t^j}^{MCI} = \frac{c\sigma_{X_{t-\tau}^i}}{\sqrt{\sigma_{X_t^j}^2 + c^2 \sigma_{X_{t-\tau}^i}^2}} \tag{3}$$

where $\sigma_{X_{t-\tau}^i, X_t^j}$ are the variances of the noise/innovation terms driving $X_{t-\tau}^i$ and $X_t^j$, respectively, and $c$ is their coupling coefficient. In practice, also non-linear links can often be well detected with ParCorr in so far they be linearly approximated. In case the linear part is even "stronger" than the non-linear part, ParCorr might also have a better detection power than a non-linear independence test (Runge et al., 2018).

## 2.6 Data

### 2.6.1 Artificial Dataset - Test Model

We tested the algorithm on artificial datasets prior to its application to real world data. The artificial dataset was created using a test model which takes time series of measured global radiation ($\mathcal{R}g$) and creates three artificial time series that conceptually
5   represent temperature ($\mathcal{T}$), gross primary production ($\mathcal{GPP}$) and ecosystem respiration ($\mathcal{R}eco$). Note that this test model is not intended to accurately represent observed land-atmosphere fluxes, but only serves to test the procedure. The model incorporates one linear auto dependence $\mathcal{T}_{t-\tau_1} \to \mathcal{T}_t$, one linear additive cross-dependence $\mathcal{R}g_{t-\tau_2} \to \mathcal{T}_t$ and two non-linear dependencies, multiplicative $\mathcal{R}g_{t-\tau_3} \bullet \mathcal{T}_{t-\tau_4} \to \mathcal{GPP}_t$, and multiplicative exponential $\mathcal{GPP}_{t-\tau_5} \bullet c^{\mathcal{T}_{t-\tau_6}} \to \mathcal{R}eco_t$ (cf. Fig. 1) according to the equations:

$$\mathcal{R}g_{mo} = \mathcal{R}g_{obs} \tag{4}$$

$$\mathcal{T}_{mo}(t) = c_1\, \mathcal{T}_{mo}(t-\tau_1) + c_2\, \mathcal{R}g_{mo}(t-\tau_2) + \xi_T \tag{5}$$

$$\mathcal{GPP}_{mo}(t) = c_3\, \mathcal{R}g_{mo}(t-\tau_3) * \mathcal{T}_{mo}(t-\tau_4) + \xi_{GPP} \tag{6}$$

$$\mathcal{R}eco_{mo}(t) = c_4\, \mathcal{GPP}_{mo}(t-\tau_5) * c_5^{\frac{\mathcal{T}_{mo}(t-\tau_6)-T_{ref}}{10}} + \xi_{Reco} \tag{7}$$

The parameters $c_1, c_2, ..., c_5$ are referred to as coupling coefficients, and the time lags are noted as $\tau_1, \tau_2,..., \tau_5$. The subscripts
$_{mo}$ and $_{obs}$ abbreviate model and observation, respectively. $T_{ref}$ is set to 15°C. The term $\xi$, termed "intrinsic" or "dynamical noise", here represents values from uncorrelated, normally distributed noise. Having dynamic noise is essential for a method utilizing conditional independence tests. It is based on the assumption that a process or state is never fully described by its deterministic part because there are unresolved intrinsic processes, summarized as $\xi$.

The model was fitted to real observational data (using radiation, temperature and land-atmosphere fluxes) of daily time
resolution, measured by the eddy-covariance method (Baldocchi et al., 1988; Baldocchi, 2003) from FLUXNET, by minimizing the sum of squared residuals using the gradient descent implemented in the `Optim.jl` package (Mogensen and Riseth, 2018). We fitted the model to 72 sites listed in Table B1 given in the Supplementary Material section. The value range for the coupling coefficients $c_1$ to $c_4$ and $c_5$ were set to [0.2,1] and [1,2.5], respectively. The lags were limited to integer values in the range [0,25]. The distributions of obtained lags and coupling coefficients are given in the Supplementary Material Fig. B1 to E1. The
fitting process thus generated 72 sets of parameters, containing coupling coefficients and lag values, which were used for the time series generation.

From each of the 72 sets of parameters we generated four sets of time series each having a length of 500 years. The time series generation was initiated using two types of data: first, uncorrelated, normally distributed noise, and second, unprocessed radiation data as used during the fitting (the available radiation data was repeated to 500 years). The resulting datasets are
called baseline dataset and seasonality dataset, respectively. In both cases, the model was run twice, once with homoscedastic (constant variance of $\xi$), once with heteroscedastic dynamical noise $\xi$. To induce heteroscedasticity, $\xi$ was multiplied with a

mean daily variance that was extracted for each variable at each FLUXNET site. In Supplementary Material Fig. F1 a five year time series excerpt from Hainich site (Knohl et al., 2003b) is shown. A third dataset is generated by anomalization (subtraction of smoothed seasonal mean) of the seasonality dataset.

### 2.6.2 Eddy Covariance Data - Majadas de Tiètar Experimental site

Data from three towers located in Majadas de Tiètar, (ES-LMa, ES-LM1, ES-LM2), a Mediterranean Savanna in central Spain, are used (coordinates of central tower: 39°56'25"N 5°46'29"W). Meaurements include the exchange of $CO_2$ between the land surface and atmosphere at half-hourly resolution using the eddy covariance method. The three tower footprints received different fertilisation treatments in spring 2015 (El-Madany et al., 2018). We consider data from before the fertilization from April 2014 to March 2015 of shortwave downward radiation (Rg), air temperature (T), net ecosystem exchange (NEE), vapour

pressure deficit (VPD), sensible heat (H) and latent heat (LE). The average temperature within this period was 17.3°C, with a total precipitation of 765mm. Most precipitation fell between October and April.

    We expect the causal imprints in the data to vary between seasons and during the course of the day. To satisfy causal stationarity, we estimate networks separately for each month and consider only samples for which the potential radiation was above $\frac{4}{5}$ of the potential daily maximum, which corresponds to midday samples. We used a mask type that limits only the

receiver variable to the respective month and day time values (cf. Table A1 for PCMCI parameter settings). This setting causes time series lengths ranging from 239 datapoints in December to 372 datapoints in July. Minimal and maximal lags were set to 0 and 8, respectively. We left the data unprocessed, i.e. we did not subtract a seasonal mean for anomalisation. Constraining the samples to separate month and midday values reduces the effect of seasonality as a common driver that would lead to spurious links. Furthermore, to correct for multiple testing we applied the Benjamini-Hochberg false discovery rate correction

(Benjamini and Hochberg, 1995). Thereby, the *p*-values for the whole graph obtained from the MCI step are adjusted to control the number of false discoveries (Runge et al., 2018). We chose a two-sided significance level of 0.01.

### 2.6.3 Gridded global data set

The second observational case study was performed on a global data set. We used data with 0.5° spatial and monthly temporal resolution from 1982 to 2008. The dataset is composed of three climatic variables, global radiation (Rg), temperature (T) and

precipitation (P), and one vegetation state index, the Normalized Difference Vegetation Index (NDVI). Both temperature and precipitation datasets were taken from the Climate Research Unit (CRU), version TS3.10 (Harris et al., 2014). The radiation data stems from the Climate Research Unit and National Centers for Environmental Prediction dataset (CRUNCEP, Viovy, 2016). The used NDVI data stems from the Global Inventory Monitoring and Modeling Systems (GIMMS) in version 3g_v1 (Pinzon and Tucker, 2014).

To examine the influence of radiation, temperature, and precipitation on NDVI by means of PCMCI we used the following settings. We compute the anomalies by subtracting a smoothed seasonal mean. A maximal time lag of three months was chosen based on the largest lag with significant partial correlation among all pairs of variables, partialling out only the autocorrelation of each variable. The receiver variable was limited to the growing season defined by T>0 and NDVI>0.2, which allows good

comparison to Wu et al. (2015). The significance level ($\alpha_{pc}$) in the condition selection phase (cf. Sect. 2.4) was chosen based on the AIC selection criterion. A concise list of PCMCI parameters that were altered from default settings is given in Table A1.

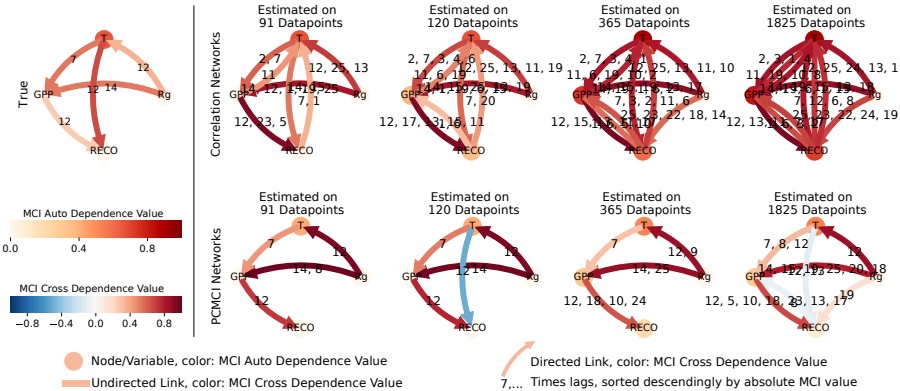

**Figure 1.** The artificial datasets are generated with a prescribed interaction structure (True Network), which is obtained by fitting the test model to the FLUXNET sites. Here we show for four time series lengths the process graphs estimated via both lagged correlation and PCMCI. The data used stems from the homoscedastic realisation of the seasonality dataset of the Hainich site. The significance level was set to 0.01. The number of time lag labels were limited to five in the correlation networks. But for the longest time series typically the whole range of lags (0-25) was significant.

## 3 Results

### 3.1 Test Model

As motivating example, in Fig. 1 we show PCMCI and lagged correlation networks in the form of process graphs. It is clearly visible, that many more spurious links pass the significance threshold of 0.01 using lagged correlation as compared to using

PCMCI. Those spurious links can complicate the analysis or lead to false conclusions and misleading hypotheses. We examined four cases of different time series lengths: 91 [183<=doy<=274] and 120 [153<=doy<=274] days, 1 and 5 years for daily data (doy: day of the year). For each time series length and each parameter set, the causal network structure was estimated for 100 realisations of the model (each based on a realization of intrinsic noise), which allowed the estimation of false positive (FPR) and true positive (TPR) detection rates. The detection rates are calculated for each tower, FPR in general and TPR link-wise.

The TPR for each link is its sum of detections among 100 realisations divided by 100. The FPR is the number of falsely detected links divided by the number of all possible false links and 100. The summary of the experiments i.e. the overall false positive rate (FPR) and the distributions of the link's true positive rate (TPR) across sites are given in Fig. 2 and 3, respectively. The blue violin plots always report the case of normal distributed (non heteroscedastic) intrinsic noise and the corresponding orange violin plots summarize the case of heteroscedastic noise. The effect of heteroscedasticity and seasonality is then assessed by

comparing the distributions obtained from the baseline dataset to the results of the seasonality dataset.

The FPR of homoscedastic time series in the baseline dataset is in the expected range of 0.01, the chosen significance level, indicating a well calibrated test due to fulfilled assumptions. The assumption of stationarity is violated as soon as heteroscedas-

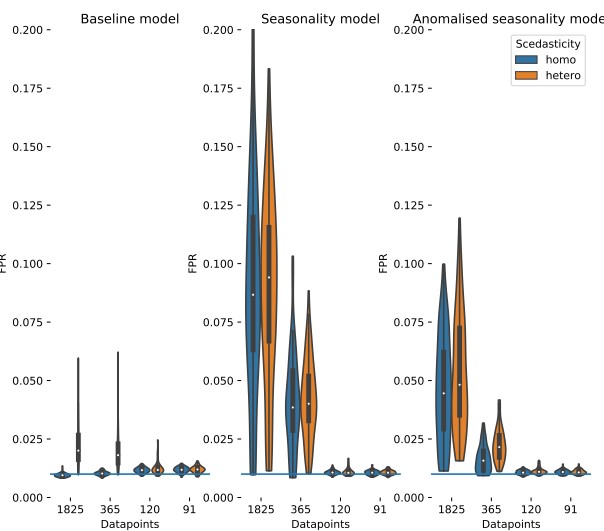

**Figure 2.** The distribution of false positive detection rates estimated for the baseline dataset, the seasonality dataset and the anomalised seasonality dataset (mean seasonal cycle subtracted). The distributions are given for different time series length (number of datapoints). Additionally, the distributions are split to show the impact of heteroscedastic noise (orange) compared to normal distributed noise (blue). The significance level of 0.01 is given by a blue horizontal line.

ticity or seasonality is present. The effect on the FPR is an increase above 0.01 for time series length of 1 and 5 years with a much stronger increase due to seasonality (factor of 4 and 8, respectively) than for heteroscedasticity (factor of 2).

The effect of non-stationarities on the TPR differs among the links. The detection of linear links ($\mathcal{R}g \to \mathcal{T}$ and $\mathcal{T} \to \mathcal{T}$) is not affected by seasonality and slightly improves for heteroscedastic dynamical noise. The detection of non-linear links is
improved by seasonality with the strongest effects in the link $\mathcal{T} \to \mathcal{GPP}$. The link $\mathcal{T} \to \mathcal{Reco}$ has a stronger non-linearity and therefore the detection rate shows a weaker effect on seasonality. Furthermore, the coupling coefficient $c_5$, the base of T, can be close to or be exactly one (cf. Fig. C1). This would actually cause on the one hand the effect of $\mathcal{T} \to \mathcal{Reco}$ to vanish, rendering a detection impossible and on the other hand result in a linear dependence of $\mathcal{GPP}$ on $\mathcal{Reco}$ which improves its detection. Heteroscedasticity seems to have a slight negative effect on non-linear links. In general, the TPRs in the seasonality dataset are
quite high, even for non-linear links, and predominantly above 80% and often reaching 100%.

Comparing the TPRs of the non-linear links shows some disparity. The links $\mathcal{T} \to \mathcal{GPP}$ and $\mathcal{T} \to \mathcal{Reco}$ experience zero detection in the baseline dataset but partially considerable rates in the seasonality dataset with a strong dependence on the time series length. On the contrary, the median of the TPR of the links $\mathcal{R}g \to \mathcal{GPP}$ and $\mathcal{GPP} \to \mathcal{Reco}$ is above 95% in the seasonality dataset, even for time series length as short as 91 days, but remains high in the baseline dataset. The removal of
the seasonal cycle keeps the TPRs largely unaffected, but reduces the FPR. Nevertheless, it still remains above the significance level by a factor of four and two for five and one year time series length, respectively.

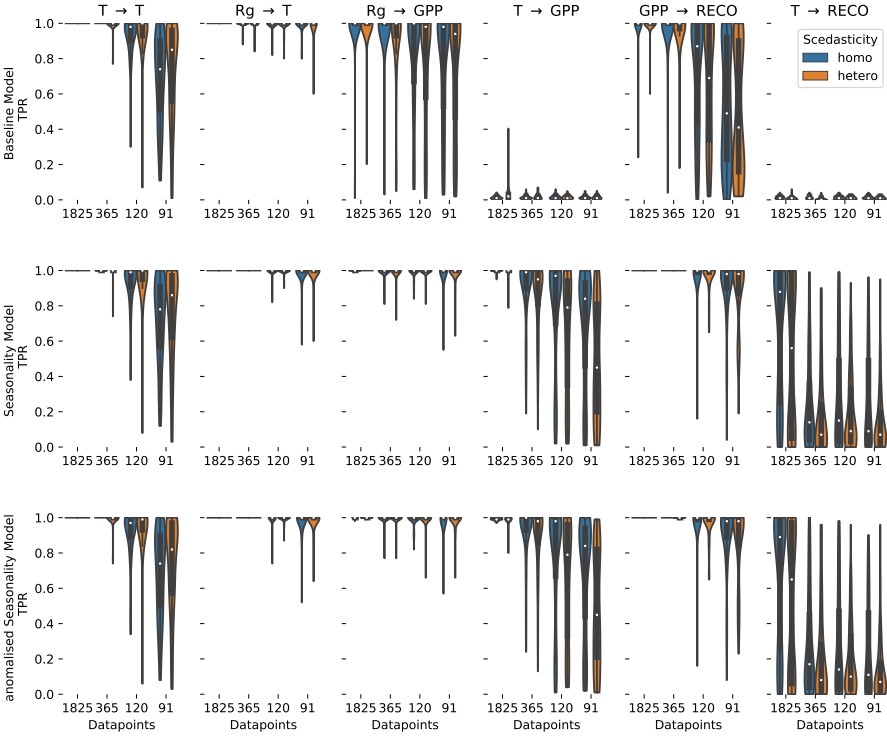

**Figure 3.** The distribution of the true positive detection rates for each link in the test model estimated for the baseline dataset, the seasonality dataset, and the anomalised seasonality dataset. The distributions are given for different time series length (number of datapoints). Additionally the distributions are split to show the impact of heteroscedastic noise (orange) compared to normally distributed noise (blue).

In summary, the seasonality dataset exhibits high TPR even for non-linear links. Compared to stationary time series, the detection of non-linear links actually benefits from seasonality. The high detection, though, comes at the cost of a high false positive rate for time series length of and above one year. To a certain degree, the increase in FPR can be counteracted by anomalization.

## 3.2  Majadas de Tiètar dataset

At first we look at the link consistency by comparing networks that were obtained for each tower within a month. The comparison is done for two months with strongly differing climate conditions: April and August. In Fig. 4 we compare the estimated link strengths (effect size estimated via partial correlation) as long as the corresponding links are significant in at least one network. The confidence intervals are overlapping for the majority of links, suggesting that the uncertainty of the fluxes is much smaller than the observed effects (El-Madany et al., 2018). Exceptions are found for only a few links (Rg$\overset{0}{\rightarrow}$T, Rg$\overset{1}{\rightarrow}$LE, VPD$\overset{1}{\rightarrow}$VPD, H$\overset{2}{\rightarrow}$H, NEE$\overset{0}{\rightarrow}$LE, H$\overset{2}{\rightarrow}$NEE, NEE$\overset{4}{\rightarrow}$NEE, number above the arrow indicates the lag) where the detection rates do not or barely overlap. Cross links (a link from one variable to another) with two or more significant appearances are predomi-

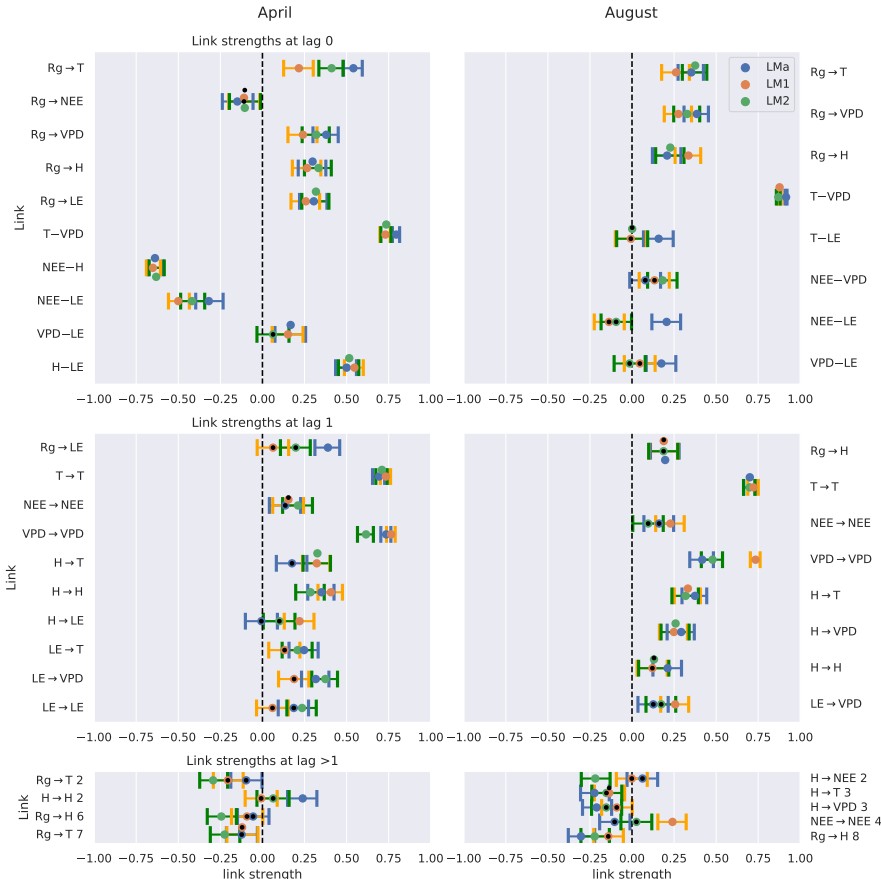

**Figure 4.** Comparison of the networks of three eddy covariance measurement stations (LMa, LM1, LM2) located in Majadas (Spain). Links that are found to be significant in one of the three networks are included. For each link, the calculated strength of all three networks is plotted with its 90% confidence interval. The colors blue, orange, and green correspond to the towers LMa, LM1, and LM2, respectively. The significance threshold is 0.01. If a link does not pass the significance, it is marked by a black dot. The links are grouped into lag 0 (top), lag 1 (middle) and all lags greater than 1 (bottom). Negative NEE is associated with carbon uptake by the ecosystem. Links at lag 0 are left undirected (−), yet as Rg is set as main driver, links incorporating Rg at lag 0 are directed (→).

nantly at zero lag. Approximately half of the links with lag one are auto-dependent links (a link from the past of a variable to its present). Comparing the links between the month April and August, distinct differences can be noticed. First, August has slightly fewer significant links compared to April. Second, the only links remaining that are significant in two or three towers are between atmospheric variables. Third, the remaining link strengths tend to be weaker in August than in April.

The difference among the seasons is further investigated in Fig. 5 which shows process graphs for each month of the year. We combine the networks of the three towers to one process graph by plotting every link that is significant in at least one tower. The process graphs in Fig. 5 visualize clearly gradual changes within the interaction structure of the biosphere-atmosphere system during the course of a year. During the main growing season from February to May, NEE is coupled strongly to the energy fluxes latent (LE) and sensible heat (H). These connections weaken, disappear or even switch sign with start and course of the dry season. Less regularly NEE also shows connections to radiation (Rg) and temperature (T). Between the atmospheric variables, a basic network between VPD, T, Rg and H remains intact and relatively constant in strength. The dominance of contemporaneous links is found as well, as seen already in Fig. 4. Besides the decoupling of NEE from any variable in the dry period, there are additional interesting patterns. For example, the positive reappearance of the link between NEE and LE in September. Here, the onset of precipitation events (cf. Fig. J1) occurred that lead to strong respiration peaks (Ma et al., 2012). Creating such a network via lagged correlation would result in much more significant links (causing the network to be not interpretable as opposed to PCMCI) and NEE does not decouple from the atmosphere in August (cf. Fig. G1).

The above results demonstrate that PCMCI is sensitive enough to capture seasonal differences and certain physiological reasonable biosphere behaviour. Moreover, PCMCI yields a better interpretable network structure than pure correlation approaches.

## 3.3 Gridded global data set

Subject of inspection were the significant lags and MCI values of each climatic variable on NDVI. In Fig. 6 the maximal MCI value and the corresponding lag are plotted for the links $X_{t-\tau} \rightarrow NDVI : \tau \in \{0,1,2,3\}$ with X being one of the climatic drivers radiation, temperature, or precipitation. The chosen significance threshold was set to 0.05. Fig. 7 shows the climatic driver with largest MCI per grid point. PCMCI detects a regionally varying influence of climatic drivers. As expected, the boreal regions are strongly driven by temperature instantaneously, while (semi-) arid regions, which correspond predominantly to grass or prairie dominated areas, respond strongest to precipitation at a time lag of one month. Radiation is found to have a comparatively low spatial effect with hot spots in south and east China, central Russia and east Canada.

The dominant lags are found to be zero and one. Just a very small fraction of the total area shows a maximal MCI value at a higher lag of two or three months. The lags are also not equally distributed among the climatic drivers. Radiation and temperature are predominantly strongest at lag zero, while precipitation has a much larger fraction of area showing the strongest response at lag one. Regions where the impact of Rg on NDVI is strongest at lag 1 tend to respond negatively to Rg but positively to precipitation at lag one. On the other hand, a large part of regions with the strongest impact of precipitation at lag zero respond negatively to it but positively to radiation.

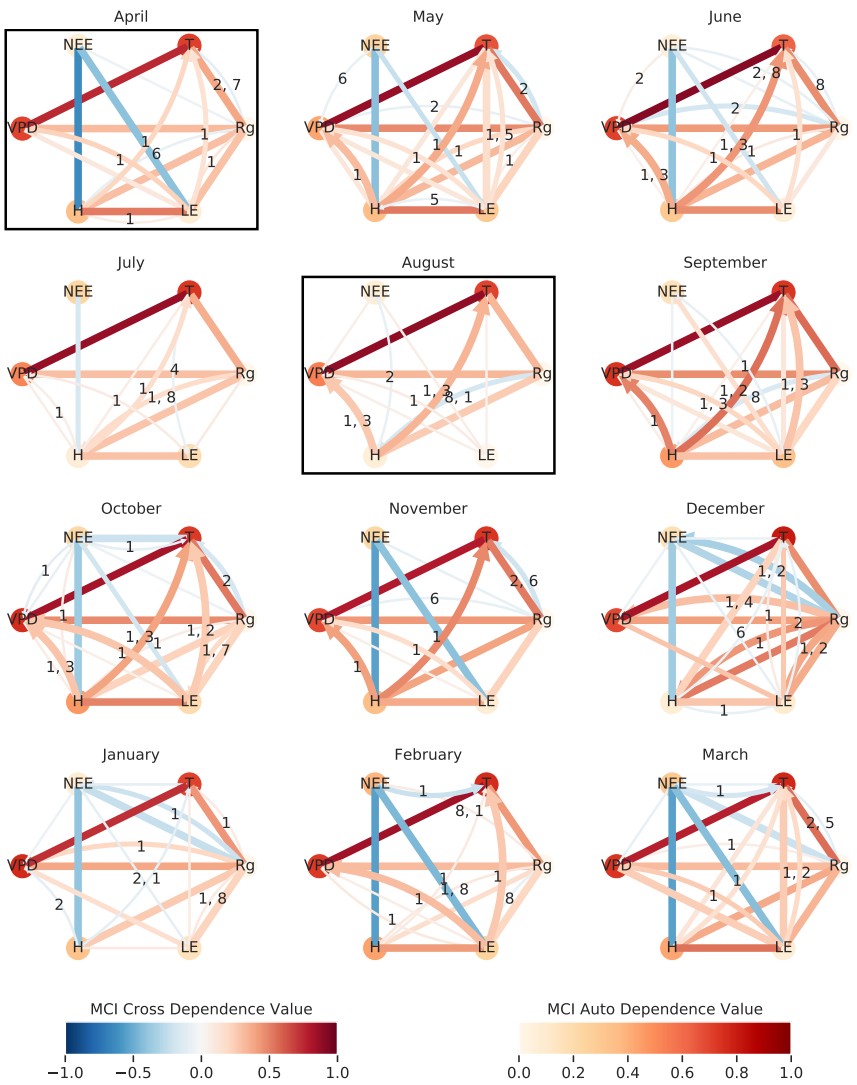

**Figure 5.** To visualize the gradual changes in interaction structure the networks of the three towers are combined for each month. The number of significant occurences of a link is given by its width. The link strength, given by the link color, is calculated by averaging the significant links of the towers. The link's lag is shown in the centre of each arrow, sorted in descending order of link strength. The resulting graphs are shown for April 2014 till March 2015. The significance threshold is 0.01. The networks of April and August, illustrated in Fig. 4, are highlighted by a box.

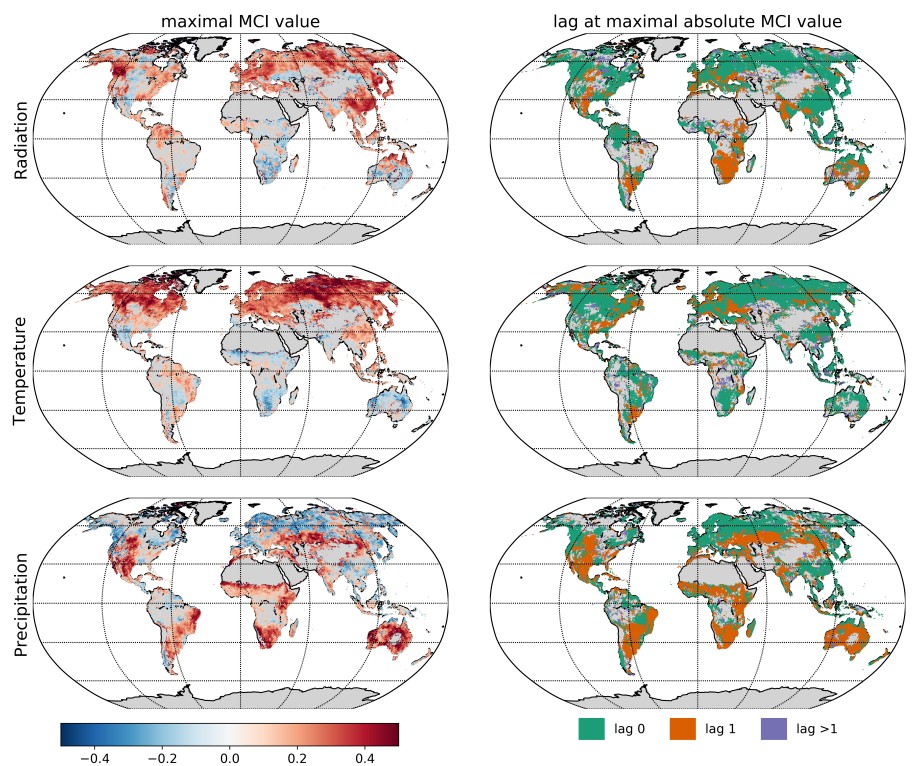

**Figure 6.** Influence of climatic drivers on NDVI as calculated by PCMCI. The first column shows the estimated causal influence given as maximal absolute MCI value of climatic drivers on NDVI. The second column gives the time lag at which the maximal absolute MCI value occures (in month).

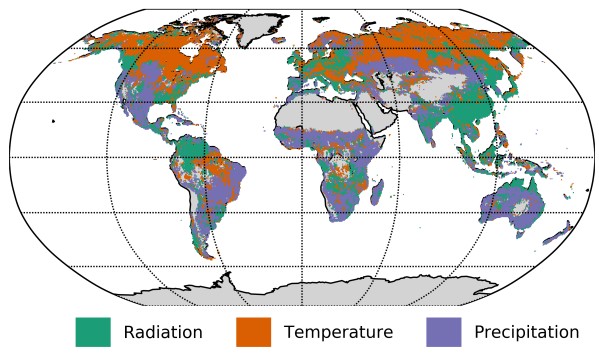

**Figure 7.** Map of the strongest climatic driver (largest absolute MCI value) per grid point.

In summary, PCMCI estimates coherent interaction patterns which match well with anticipated behaviour based on vegetation type and prevailing climatic conditions.

## 4 Discussion

Causal discovery methods promise an improved understanding and can help to come up with new hypotheses about the interaction between biosphere and atmosphere (Christiansen and Peters, 2018; Runge et al., 2019a). But the underlying assumptions need to be properly taken into account. The coupled biosphere-atmosphere system possesses several challenges that potentially violate the underlying assumptions of causal discovery in general and the employed method's assumptions in particular. Here, we investigate the effect of a violation of assumptions on PCMCI network estimates.

With regard to expected non-linearities in biosphere-atmosphere interactions, using a linear independence test within the PCMCI framework may not be adequate. We motivate our choice with the following arguments: first, non-linearities are often approximated linearly. Second, a linear regression based test has a much higher power for detecting linear links than a non parametric test (Runge, 2018a) and can, hence, detect links already at smaller sample sizes. Third, linear partial correlation is easily interpretable, for example, positive and negative MCI values. This motivation is supported on the one hand by the results of the test model and on the other hand by additionally performed analyses on the observational datasets using Gaussian regression and distance correlation as an independence test. These results (cf. Fig. H1,I1,K1) show similar patterns but due to the low sample sizes exhibit lower statistical significances. In general, the derived results show a high detection power with a strong consistency in calculated effect strengths on eddy covariance data and global, regularly gridded reanalysis data that leads to well interpretable patterns. Observed drawbacks are a high FPR in case of violated assumptions, especially strong periodicity, as well as the appearance of contemporaneous lags in measurement datasets.

### 4.1 Lessons learned from the test model

The probability to detect a link with PCMCI depends strongly on a link's MCI effect size, which is larger for strong variance in the driver and a low variance in the receiver (cf. Sect. 2.5). Several results can be explained by this observation. First, the variance of three out of five drivers of cross dependencies in the test model are either directly or indirectly (via $\mathcal{GPP}$) influenced by $\mathcal{R}g$, which has the highest variance of all variables. Consequently, the detection power of the three links is large, almost 100%. In comparison, the other variables' variances are weaker, since they are influenced by $\mathcal{T}$, which results in a lower detection power. This is the origin of the disparity in detection rates of the non-linear links. Second, also the partially strong increase in TPR of non-linear links (influenced in a multiplicative way by $\mathcal{T}$) from the baseline dataset to the seasonality dataset can be explained by this increase in variance. A multiplicative link is actually not generally expected to be found by ParCorr (Runge, 2018a), but the value of the multiplicative factor is dominated by the seasonal value, and not the dynamical noise, which might cause rather a scaling of the dynamical noise terms rather than a random distortion. Third, the dependence on the variance ratio can also explain the difference in TPR between homoscedastic (equal error variance) and heteroscedastic (error variance changing over time) time series, i.e., the variance of $\mathcal{R}g$ and $\mathcal{GPP}$ exhibits a strong seasonality with its peak in summer, while the variance of $\mathcal{T}$ is rather constant. This explains, for example, the strong decrease in TPR for the link $\mathcal{T} \rightarrow \mathcal{GPP}$ at 91 days time series length when comparing homoscedasticity to heteroscedasticity. The decrease in TPR is less pronounced when another season, implying a different variance, is chosen for this comparison. As links with weak driver

variance and strong response variance are more likely to be missed, one may ask which effect this will have on the detection of feedback loops where one variable has low and the other high variance. Here lies a limitation of the test model where no feedback loops were implemented.

Seasonality and heteroscedasticity constitute violations of the stationarity assumption underlying the independence test ParCorr. Seasonality constitutes a common driver in this model. In general, such common drivers increase the dependence among the variables and hence, lead to a higher detection rate for true links (TPR) as well as a higher false positive rate (FPR) for absent links if this driver is not conditioned out properly. This additionally causes the TPR and the FPR rate to increase in the seasonality model. As shown in Runge (2018a), including the cause of the non-stationarity as an exogenous driver in the analysis allows PCMCI to regress out its influence on the other variables. However, for ParCorr this is only valid if the dependence on the non-stationary driver is linear. Therefore, the regression on $\mathcal{R}g$ fails for $\mathcal{GPP}$ and $\mathcal{R}eco$ in the test model. With this ill-posed setting, the probability to detect false links increases with increasing time series length or when more periods are included. Stationarity in mean is obviously also not fully guaranteed when subtracting the seasonal mean. Here we observe that the FPR stays above the significance level for the anomalised seasonality dataset. One can ask whether the FPR stays above the significance threshold because subtracting the seasonal mean does not remove the heteroscedasticity. However, we attribute this high FPR to a not fully removed seasonality since the FPR of both homoscedastic and heteroscedastic time series decreases by roughly the same amount in the anomalised seasonality dataset and the effect of heteroscedasticity is rather weak in the baseline dataset. The increasing FPR with increasing time series length can further raise doubts regarding the analysis of long time series. For such an analysis, though, the assumption of causal stationarity should first be assessed. For example, the link from radiation to GPP vanishes in winter as there is mostly no active plant material left. To account for causal stationarity, the analysis should be limited to time series sections where the causal structure is expected to be similar. This is typically done by limiting the analysis to a specific time period (i.e. 'masking'), e.g. a specific season, month, or time of the day. Such masking reduces additionally further influences of remaining seasonality or heteroscedasticity. One can argue, as it is done in Peters et al. (2017), that the causality of a system is invariant even between seasons because the physical mechanism is the same in all seasons. Yet, while the physical, i.e. functional relationship might be constant over time, its imprint in the time series might vary. For example, a functional dependence $f(x)$ might be 'flat' for small values of $x$ and linearly increasing for larger values. If only small values occur in the winter season, then the link is absent, while it 'appears' only in the summer season. Across all seasons, this can be considered as a nonlinear functional dependence $f(x)$. In practice, restricting an analysis to different seasons can help in interpreting the mechanism, here in a linear framework.

Summarizing the results of the test model, the different detection rates, disparity among non-linear links, and the detection of multiplicative links are largely explainable via the effect of the variance on the link detection. Yet, the discussion revealed the need for further research in several aspects. On the one hand, feedback loops are not included in the test model yet are an important aspect in natural systems. On the other hand, removing non-stationarities is essential to keep the false positive rate in the expected range, but standard procedures of subtracting the mean seasonal cycle are not sufficient. Further, the effect of non-stationarity on the causal network structure needs to be investigated.

## 4.2 Causal interpretation of estimated networks from observational data

In both the half-hourly time resolved eddy covariance data and the monthly global dataset the predominant type of dependence found is contemporaneous. PCMCI leaves these undirected since no time order indicating the flow of causal information is available. Further, as discussed in Sect. 2.2, contemporaneous common drivers or mediators are not accounted for. The consequence is that both spurious contemporaneous and spurious lagged links can appear, if they are due to contemporaneous variables. For interactions that are contemporaneous in nature since they occur on considerably shorter time scales than the time resolution, therefore, PCMCI is not the optimal choice regarding a causal interpretation and other methods should be applied in conjunction (Runge et al., 2019a). Further, we faced a trade off between fulfilling causal assumptions and detection power. In practice, accounting for causal stationarity (by limiting the analysis to certain periods of the dataset) means decreasing the number of available data points while accounting for causal sufficiency leads to an increase in dimensionality by adding variables and increasing the maximal lag. Both will lead to a decrease in detection power, which can affect the network structure. PCMCI alleviates the curse of dimensionality by applying a condition selection step, but still one cannot indefinitely add more variables. Another important factor that affects detection power and dimensionality is the time resolution. There are several points in favour and against increasing time resolution. On the one hand, increasing time resolution can resolve contemporaneous links and potentially increases the detection power due to an increased number of datapoints. On the other hand, the dimensionality increases if the maximal lag is adapted. Further, causal information might be split apart and distributed over more lags, rendering the links at each individual lag less detectable. This can cause links to disappear, but links can also appear if new processes are resolved at a higher time scale. At last, observational noise (measurement errors) might be larger in higher resolution data than in aggregated data, as it is averaged out in the latter and thus affects link detection less. Consequently, when comparing network structures based on different settings, i.e. maximal lag, included variables, time resolution, and considered time period, the (dis-)appearance of single links among specific variables can stem from several factors, i.e. a change in detection power, a changed (conditional) dependency, or due to a common driver. These factors together with a non-zero false positive detection rate are challenging for a causal interpretation. Therefore, detected links should be interpreted with care and can give rise to new hypotheses and analyses involving further variables. Generally, a causal interpretation is more robust regarding the absence of links (cf. Sect.2.2). In particular it does not require that all common drivers are observed.

Nevertheless, robust patterns were identified in our studies that are also consistent with other studies. Furthermore, a causal analysis has the advantage of an enhanced interpretability compared to correlative approaches. First of all, we could show that the networks' estimated link strengths are consistent for observational data, even though measurement error affects the data. The dataset used was suitable for this analysis, as the measurement stations are located in a reasonably homogeneous ecosystem that shows only little spatial variation (El-Madany et al., 2018). Thus, also the interaction between biosphere and atmosphere is expected to change only marginally across space within this ecosystem. Second, the gradual changes in plant activity that are taking place in the ecosystem of Majadas throughout the year do very well emerge in the coupling strength of daytime NEE to the atmospheric variables. The observed decoupling during the dry season is in accordance with the one of a soybean field during drought conditions observed by Ruddell and Kumar (2009). The gradual changes in ecosystem activity

are not visible in a pure (lagged) correlation analysis or are only visible in color or density changes but the large number of significant links prevents any detailed interpretation on the physical mechanisms and changes thereof. The large number of significant links compared to the PCMCI networks stems solely from the absence of conditioning on common drivers or mediating variables, which often further leads to an overestimation of the link strength in correlation networks. As a result, processes, such as the decoupling of NEE during the dry period, stay hidden. To reduce the effect of confounding, often analyses utilize partial correlation (see e.g. Buermann et al., 2018). However, a partial correlation can introduce new dependencies (as opposed to removing them) if one conditions on causal effects of the variables under consideration (the 'marrying parents' effect). This issue is avoided in PCMCI by only conditioning on past variables. Additionally, PCMCI chooses only relevant variables as conditions by applying the PC condition selection step which is especially valuable in high dimensional study cases and improves detection power and computation time (Runge, 2018a).

The global study of climatic drivers of vegetation shows a general pattern of lags and dependence strengths of vegetation on climatic variables that is easily-interpretable. The boreal regions appear energy limited and especially driven by temperature (cf. Fig 6c), while the strongest dependence of (semi-)arid regions on precipitation reflects their limitation in water supply. Two recent studies performed a similar analysis. Both Wu et al. (2015) and Papagiannopoulou et al. (2017b) investigated lagged effects and dependence strengths of NDVI on precipitation, temperature and radiation. Wu et al. (2015) estimated the lags of the strongest effects via an univariate regression of the climatic drivers on NDVI and subsequently used those lags to fit a multivariate regression model of the climatic drivers on NDVI and determined their relative effects. Papagiannopoulou et al. (2017b) applied a non-linear Granger causality framework utilising a random forest predictive model; the method was presented by Papagiannopoulou et al. (2017a). We recognize that similar patterns are observed in Wu et al. (2015), but the lags at the maximal MCI value are usually lower than the one found in Wu et al. (2015), which stems from the methodical differences. Besides having used anomaly values, PCMCI regresses both NDVI and the climatic drivers on their parents before calculating the MCI value (cf. Sect. 2.5). This especially removes the influence of autocorrelation. Runge et al. (2014) shows how autocorrelation affects the correlative lag causing it to be larger for stronger autocorrelation; thereby the correlative lag may become larger than the causal lag. Therefore, according to Fig. 6b, the causal information embedded in monthly resolution is predominantly received within one month. Finding the strongest causal links at a time lag up to one month appear in agreement with Papagiannopoulou et al. (2017b). Also the spatial distribution of the strongest climatic influences compares well. But there are certain noteworthy differences which not necessarily stem from masking differences, i.e. that we took only values belonging to the growing season while Papagiannopoulou et al. (2017b) took the whole time series. First, there is little significant Granger causality of water availability found in boreal regions while there are significant negative causal dependencies detected via PCMCI. Second, NDVI in arid regions is not or barely Granger caused by radiation and temperature, but in parts shows a negative PCMCI value on those variables. There might be physiological reasons that can explain the PCMCI patterns, i.e. water logging or too high temperatures. To explain the differences though, we could identify two possible reasons. First, Papagiannopoulou et al. (2017b) masked out negative influences of radiation arguing that radiation is not negatively affecting NDVI. They found, that negative influences of Rg are usually a consequence of poor conditioning on other variables. Second, a precipitation event in boreal regions coincides with a reduction in radiation and temperature. Boreal regions usually do not

suffer from water shortages. Thus they respond stronger to the reduction of radiation and temperature than precipitation. As precipitation is coupled negatively to radiation and temperature at lag zero, the effect of precipitation on NDVI is found to be negative. Thus, the link P$\xrightarrow{-}$NDVI might be an effect of the contemporaneous common driver scheme P $\xleftarrow{-}$ Rg $\xrightarrow{+}$ NDVI and therefore would not be causal. In fact, a similar argumentation can be given for the negative impact of temperature and radiation on NDVI in arid regions.

In summary, we pointed out the need for careful interpretations in applying causal discovery methods and especially highlighted the challenges linked to the study of biosphere-atmosphere interaction via PCMCI. We demonstrated that the network structures estimated from observational data are explainable with respect to plant physiology and climatic effects. Finally, our study shows that causal methods can deliver better interpretability and a much improved process understanding in comparison to correlation and bivariate Granger causality analyses that are ambiguous to interpret since they do not account for common drivers.

## 4.3 Outlook

The preceding discussion has shed light on the merits of PCMCI as well as the challenges of applying causal discovery methods. Runge et al. (2019a) discuss further challenges and methods and give an outlook how multiple methods can be combined to alleviate limitations.

# 5 Conclusions

Here we tested PCMCI, an algorithm that estimates causal graphs from empirical time-series. We specifically explored two types of data sets that are highly relevant in biogeosciences: eddy covariance measurements of land-atmosphere fluxes and global satellite remote sensing of vegetation greenness. The causal graphs estimated from the eddy covariance data collected in a Mediterranean site confirm patterns we would expect in these ecosystems: During the dry season's plants senescence, for instance, the ecosystem's carbon cycle (NEE) decouples from meteorological variability. On the contrary during the main growing season with warm and humid conditions strong links between NEE, LE and H characterise the graph. Not only the strongly contrasting states emerge in the graph structure using the causal framework, but also the gradual transitions that relate to minor changes like the connectivity of sensible heat to temperature with progressing dryness. A purely correlative analysis, instead, is not able to resolve these patterns. PCMCI allows us to identify and focus on much fewer, but highly relevant dependencies only. Applying the approach to three replicated eddy covariance systems shows the robustness of the method to random errors in the fluxes measurements and confirm one of the assumption of eddy covariance: above a relatively homogeneous terrain the fluxes measured should be spatially invariant, and so the underlying causal relationship between climate and fluxes. The global analysis of NDVI in relation to climatic drivers confirms the known patterns of dependence strengths of vegetation on climatic variables: boreal regions are energy limited and especially driven by temperature and secondarily radiation, while in semi-arid regions vegetation dynamics are strongly dependent on water supply. However, obtained response times of vegetation to climatic variations are lower using PCMCI than correlation which can be attributed to a better treatment of the autocorrelation in the time-series and cross-relations among climate variables. Compared to merely correlative approaches, this leads to a interpretable pattern of driver-response relationships. In short, the new developments achieved in causal inference allow to gain well constrained insights on processes, that would otherwise be drowning in the correlation chaos. Therefore we hope that this study fosters usage of causal inference in analysing interactions and feedbacks of the biosphere-atmosphere system and furthermore exhibits our demand of further developments.

**Table A1.** PCMCI parameters that were used differently from default settings.

| Dataset | significance | $\alpha_{pc}$ | tau_min | tau_max | selected_variables | mask_type | fdr_method |
|---------|-------------|---------------|---------|---------|--------------------|-----------|------------|
| Test Model | 0.01 | 0.4 | 0 | 25 | [1,2,3] | 'none' | 'none' |
| Majadas Dataset | 0.01 | None | 0 | 8 | [1,2,3,4,5] | 'y' | 'fdr_bh' |
| Gridded global data set | 0.05 | None | 0 | 3 | [1,2,3] | 'y' | 'fdr_bh' |

*Code and data availability.* The eddy covariance data of the FLUXNET sites can be downloaded from the official webpage (https://fluxnet.fluxdata.org/).
CRU temperature and precipitation data is available at http://badc.nerc.ac.uk/data/cru/.
CRUNCEP radiation data can be downloaded via ftp://nacp.ornl.gov/synthesis/2009/frescati/temp/land_use_change/original/readme.htm.
The NDVI dataset is available at http://ecocast.arc.nasa.gov/data/pub/gimms/3g/.

5    The TIGRAMITE software package that includes PCMCI can be found on github https://github.com/jakobrunge/tigramite/. All other code
will be made available upon request.

| FLUXNET-ID | start year | end year | Data Reference | FLUXNET-ID | start year | end year | Data Reference |
|---|---|---|---|---|---|---|---|
| AT-Neu | 2002 | 2012 | Wohlfahrt et al. | DK-ZaH | 2000 | 2014 | Lund et al. (2012) |
| AU-Cpr | 2010 | 2014 | Meyer et al. (2015) | FI-Hyy | 1996 | 2014 | Suni et al. (2003) |
| AU-DaP | 2007 | 2013 | Beringer et al. (2011a) | FI-Sod | 2001 | 2014 | Thum et al. (2007) |
| AU-DaS | 2008 | 2014 | Hutley et al. (2011) | FR-Fon | 2005 | 2014 | Delpierre et al. (2016) |
| AU-Dry | 2008 | 2014 | Cernusak et al. (2011) | FR-LBr | 1996 | 2008 | Berbigier et al. (2001) |
| AU-How | 2001 | 2014 | Beringer et al. (2007) | FR-Pue | 2000 | 2014 | Rambal et al. (2004) |
| AU-Stp | 2008 | 2014 | Beringer et al. (2011b) | GF-Guy | 2004 | 2014 | Bonal et al. (2008) |
| AU-Tum | 2001 | 2014 | Leuning et al. (2005) | IT-BCi | 2004 | 2014 | Vitale et al. (2016) |
| BE-Lon | 2004 | 2014 | Moureaux et al. (2006) | IT-Col | 1996 | 2014 | Valentini et al. (1996) |
| BE-Vie | 1996 | 2014 | Aubinet et al. (2001) | IT-Lav | 2003 | 2014 | Marcolla et al. (2003) |
| BR-Sa3 | 2000 | 2004 | Saleska et al. (2003) | IT-MBo | 2003 | 2013 | Marcolla et al. (2011) |
| CA-Man | 1994 | 2008 | Brooks et al. (1997) | IT-Noe | 2004 | 2014 | Spano et al. |
| CA-NS2 | 2001 | 2005 | Bond-Lamberty et al. (2004) | IT-Ro1 | 2000 | 2008 | Rey et al. (2002) |
| CA-NS3 | 2001 | 2005 | Wang et al. (2002a) | IT-SRo | 1999 | 2012 | Chiesi et al. (2005) |
| CA-NS5 | 2001 | 2005 | Wang et al. (2002b) | IT-Tor | 2008 | 2014 | Galvagno et al. (2013) |
| CA-NS6 | 2001 | 2005 | Wang et al. (2002c) | JP-SMF | 2002 | 2006 | Matsumoto et al. (2008) |
| CA-Qfo | 2003 | 2010 | Chen et al. (2006) | NL-Hor | 2004 | 2011 | Jacobs et al. (2007) |
| CA-SF2 | 2001 | 2005 | Rayment and Jarvis (1999a) | RU-Fyo | 1998 | 2014 | Kurbatova et al. (2008) |
| CA-SF3 | 2001 | 2006 | Rayment and Jarvis (1999b) | US-ARM | 2003 | 2012 | Fischer et al. (2007) |
| CH-Cha | 2005 | 2014 | Merbold et al. (2014) | US-Blo | 1997 | 2007 | Schade et al. |
| CH-Dav | 1997 | 2014 | Zielis et al. (2014) | US-Ha1 | 1991 | 2012 | Wofsy et al. (1993) |
| CH-Fru | 2005 | 2014 | Imer et al. (2013) | US-Me2 | 2002 | 2014 | McDowell et al. (2004) |
| CH-Lae | 2004 | 2014 | Etzold et al. (2011) | US-Me6 | 2010 | 2014 | Ruehr et al. (2012a) |
| CH-Oe1 | 2002 | 2008 | Ammann et al. (2009) | US-MMS | 1999 | 2014 | Pryor et al. (1999) |
| CH-Oe2 | 2004 | 2014 | Dietiker et al. (2010) | US-Ne1 | 2001 | 2013 | Gitelson et al. (2003) |
| CZ-wet | 2006 | 2014 | Dušek et al. (2012) | US-Ne2 | 2001 | 2013 | Cassman et al. (2003a) |
| DE-Akm | 2009 | 2014 | Bernhofer et al. (a) | US-Ne3 | 2001 | 2013 | Cassman et al. (2003b) |
| DE-Geb | 2001 | 2014 | Anthoni et al. (2004) | US-SRG | 2008 | 2014 | Ruehr et al. (2012b) |
| DE-Gri | 2004 | 2014 | Prescher et al. (2010a) | US-SRM | 2004 | 2014 | Scott et al. (2008) |
| DE-Hai | 2000 | 2012 | Knohl et al. (2003a) | US-Ton | 2001 | 2014 | Tang et al. (2003) |
| DE-Kli | 2004 | 2014 | Prescher et al. (2010b) | US-Twt | 2009 | 2014 | Hatala et al. (2012) |
| DE-Lkb | 2009 | 2013 | Lindauer et al. (2014) | US-UMB | 2000 | 2014 | Rothstein et al. (2000) |
| DE-Obe | 2008 | 2014 | Bernhofer et al. (b) | US-UMd | 2007 | 2014 | Nave et al. (2011) |
| DE-Spw | 2010 | 2014 | Bernhofer et al. (c) | US-Var | 2000 | 2014 | Xu et al. (2004) |
| DE-Tha | 1996 | 2014 | Grünwald and Bernhofer (2007) | US-Whs | 2007 | 2014 | Scott et al. (2006) |
| DK-NuF | 2008 | 2014 | Westergaard-Nielsen et al. (2013) | US-Wkg | 2004 | 2014 | Emmerich (2003) |
| DK-Sor | 1996 | 2014 | Pilegaard et al. (2011) | | | | |

**Table B1.** List of FLUXNET sites used for the generation of artificial datasets and the time periode used.

**Situation:** A process **X** is given.

Process Graph

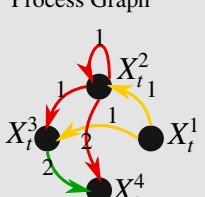

Time series Graph

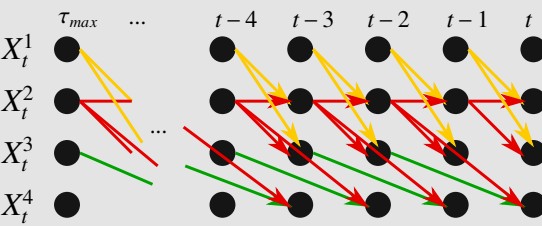

In real world study cases the dependencies of **X** are unknown.

**Aim:** To study process **X** via graphical models, we need to estimate conditional independence between all variables:

$$X_{t-\tau}^i \not\!\perp\!\!\!\perp X_t^j \mid \mathbf{X}_t^- \setminus \{X_{t-\tau}^i\}$$

**Problem:** Conditioning on the whole past $\mathbf{X}_t^-$ leads to high dimensional estimations.

**PCMCI approach**

1.

### PC-step

1. Start with a fully connected network

$$\widetilde{\mathcal{P}}(X_t^j) = \mathbf{X}_t^- = \{X_{t-\tau}^i : i = 1,...,N, \tau = 1,...,\tau_{max}\}$$

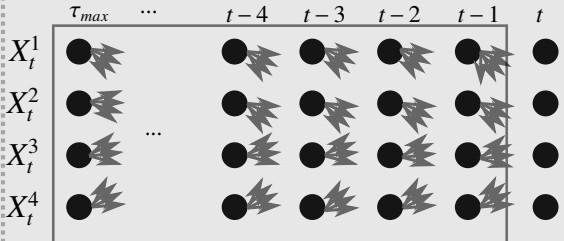

2. Remove nodes from $\widetilde{P}(X_t^j)$ iteratively by keeping nodes that are conditional dependent to $X_t^j$ at a significance level set by the parameter $\alpha_{pc}$ (typically 0.2 - 0.4). Ideally the true parents are still in the set of parents (colored arrows). But likely some spurious parents as well (grey arrows).

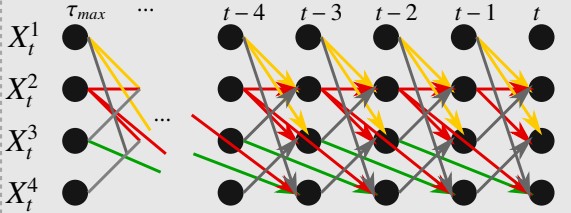

2.

### MCI-step

1. Perform the conditional independence test for each pair of nodes using their estimated set of parents from the PC-step as conditioning set.

For example a link between $X_{t-1}^2$ und $X_t^3$ exists iff:

$$X_{t-1}^2 \not\!\perp\!\!\!\perp X_t^3 \mid \widetilde{P}(X_{t-1}^2), \widetilde{P}(X_t^3) \setminus \{X_{t-1}^2\}$$

with

$$\widetilde{P}(X_{t-1}^2) = X_{t-2}^1, X_{t-2}^2, X_{t-2}^3$$
$$\widetilde{P}(X_t^3) = X_{t-1}^1, X_{t-1}^2, X_{t-1}^4$$

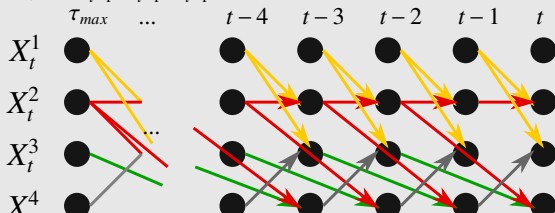

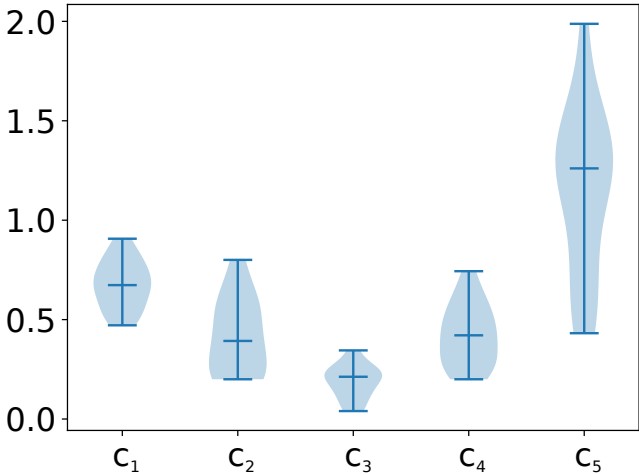

**Figure B1.** Distribution of coupling coefficients obtained after fitting the test model to the Fluxnet sites. Here shown are the distributions used for generation of heteroscedastic time series.

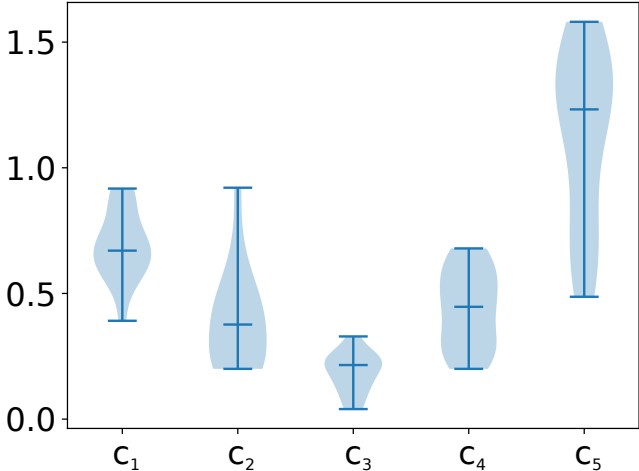

**Figure C1.** Distribution of coupling coefficients obtained after fitting the test model to the Fluxnet sites. Here shown are the distributions used for generation of homoscedastic time series.

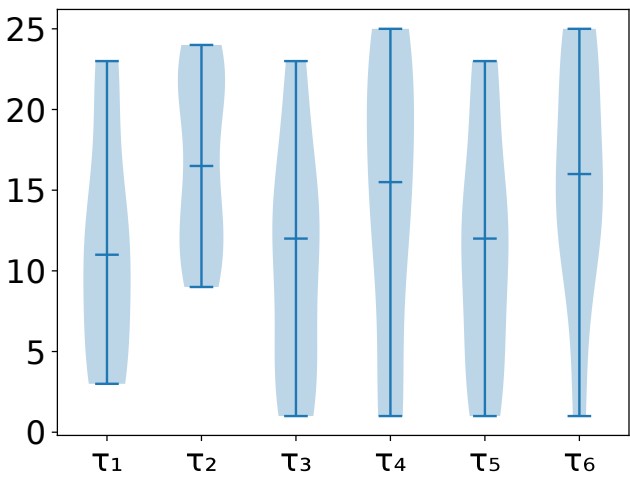

**Figure D1.** Distribution of time lags obtained after fitting the test model to the Fluxnet sites. Here shown are the distributions used for generation of heteroscedastic time series.

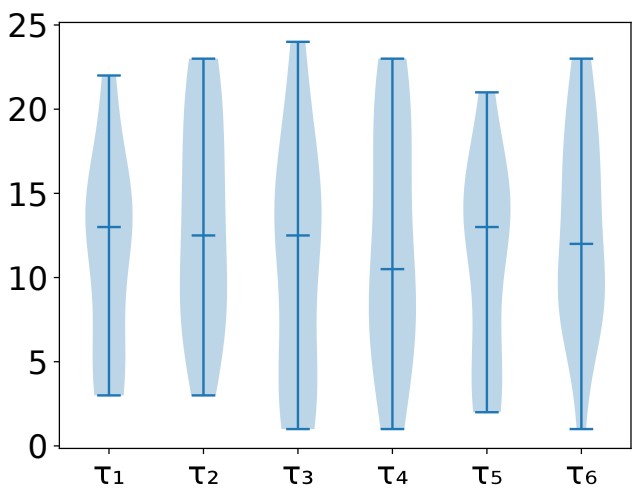

**Figure E1.** Distribution of time lags obtained after fitting the test model to the Fluxnet sites. Here shown are the distributions used for generation of homoscedastic time series.

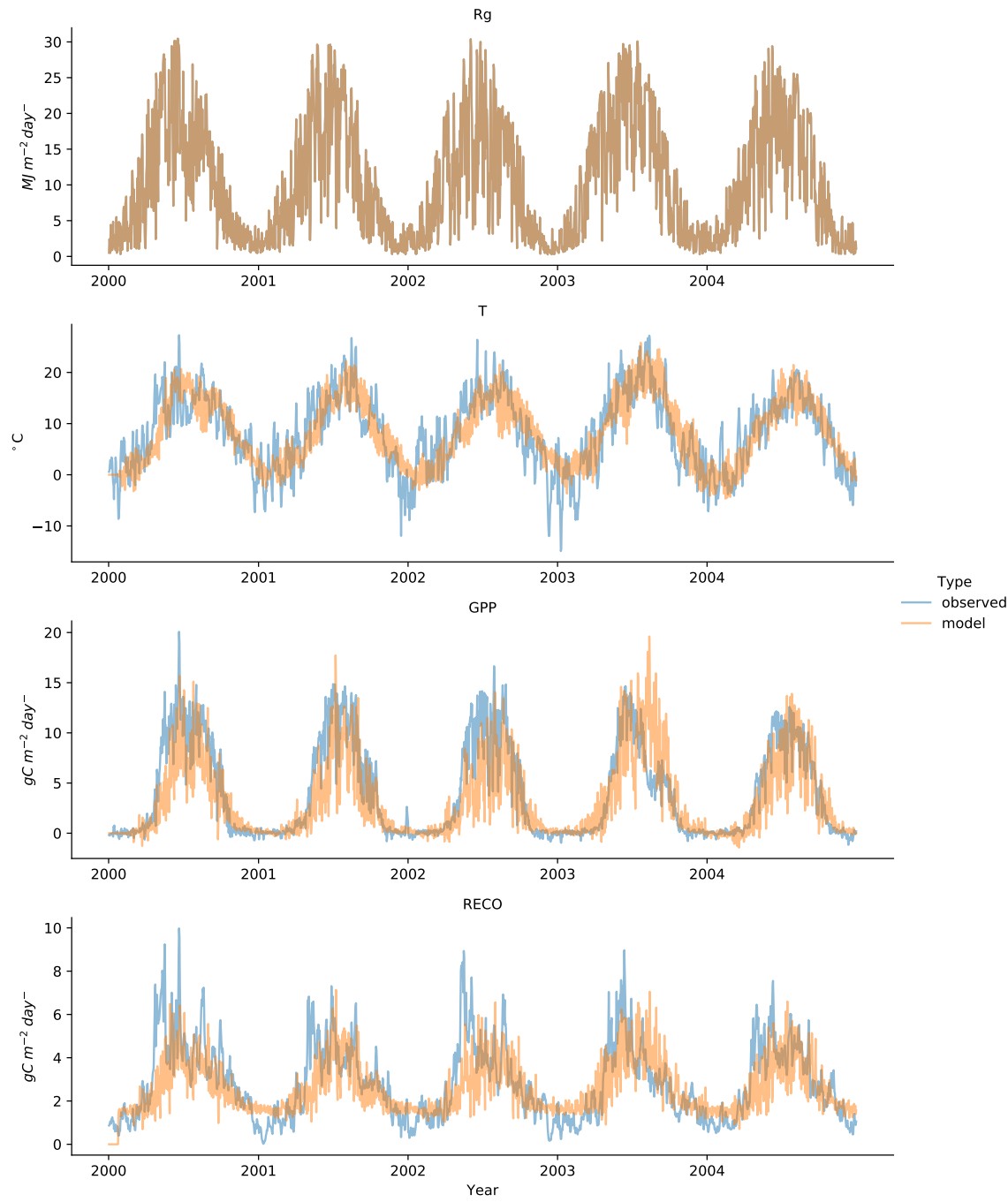

**Figure F1.** Observed (blue) and test model (orange) time-series for Hainich Fluxnet site. The model data was produced with heteroscedastic noise.

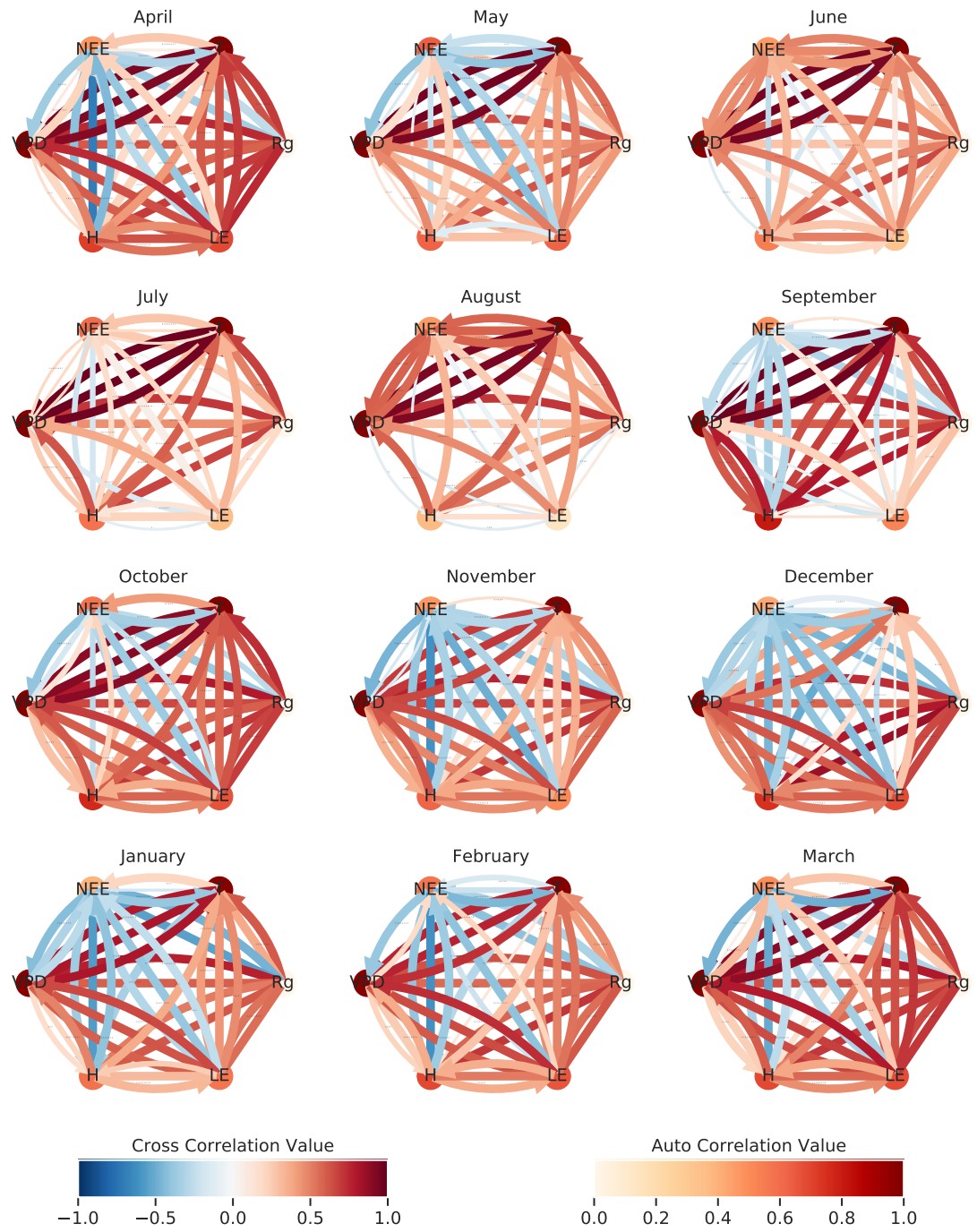

**Figure G1.** Same as Fig. 5 but using simple correlation analysis to estimate the graph structures. The number of significant occurences of a link is given by its width. The link strength, given by the link color, is calculated by averaging the significant links of the towers. Link labels indicating the lag were removed to improve link visibility. They typically ranged from 1 to 8 (full range of possible lags). The resulting graphs are shown for April 2014 till March 2015. The significance threshold is 0.01.

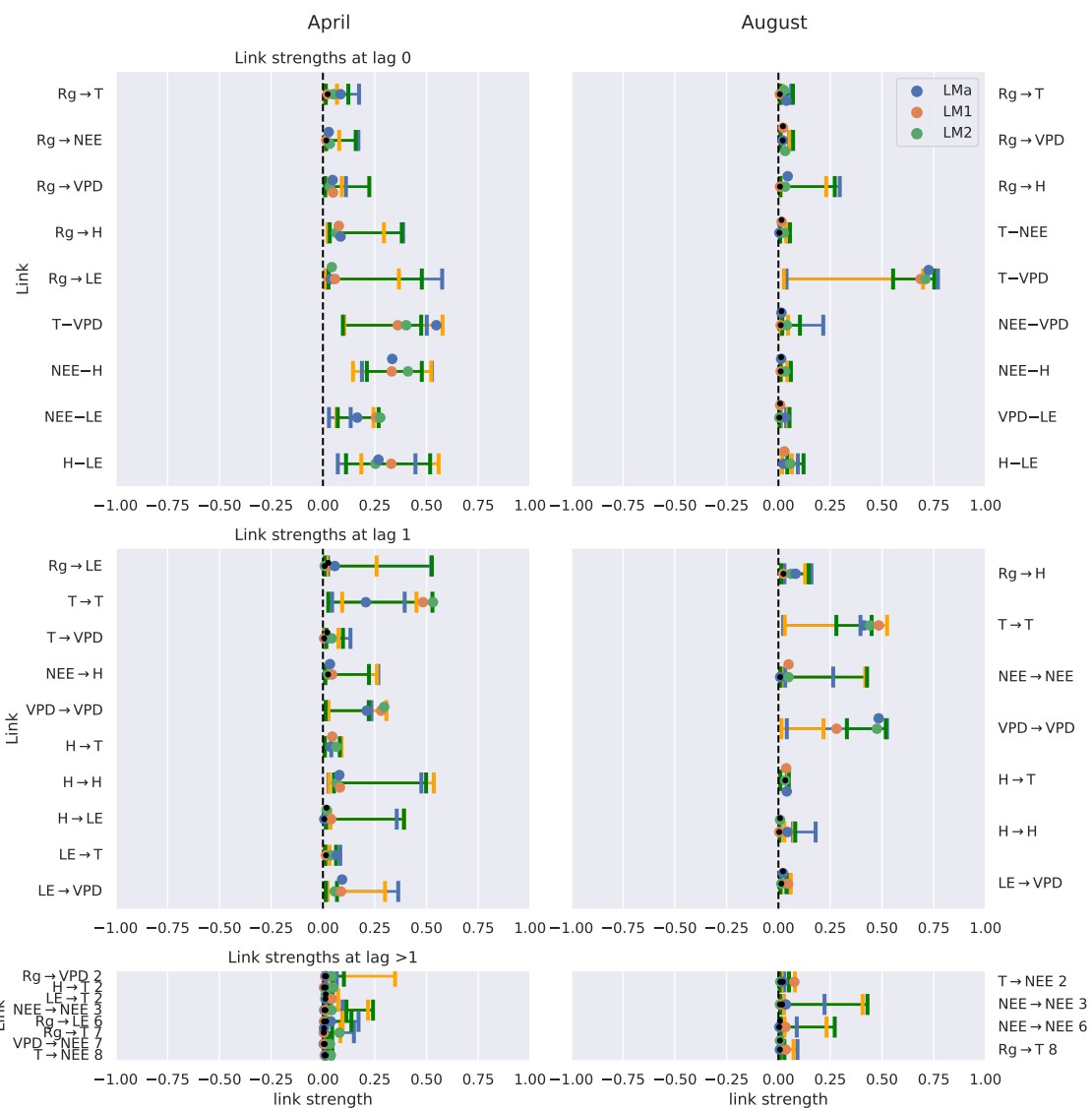

**Figure H1.** Same as Fig. 4 of the manuscript but the analysis was performed using a non-linear independence test. Comparison of the networks of three eddy covariance measurement stations (LMa, LM1, LM2) located in Majadas (Spain). Links that are found to be significant in one of the three networks are included. For each link, the calculated strength of all three networks is plotted with its 90% confidence interval. The colors blue, orange, and green correspond to the towers LMa, LM1, and LM2, respectively. The significance threshold is 0.01. If a link does not pass the significance, it is marked by a black dot. The links are grouped into lag 0 (top), lag 1 (middle) and all lags greater than 1 (bottom). Links at lag 0 are left undirected (−), yet as Rg is set as main driver, links incorporating Rg at lag 0 are directed (→). Note that GPDC only yields positive link strengths. Further, the strength values estimated with GPDC are rather weak due to the low number of datapoints and the larger sensitivity of that method to the sample size

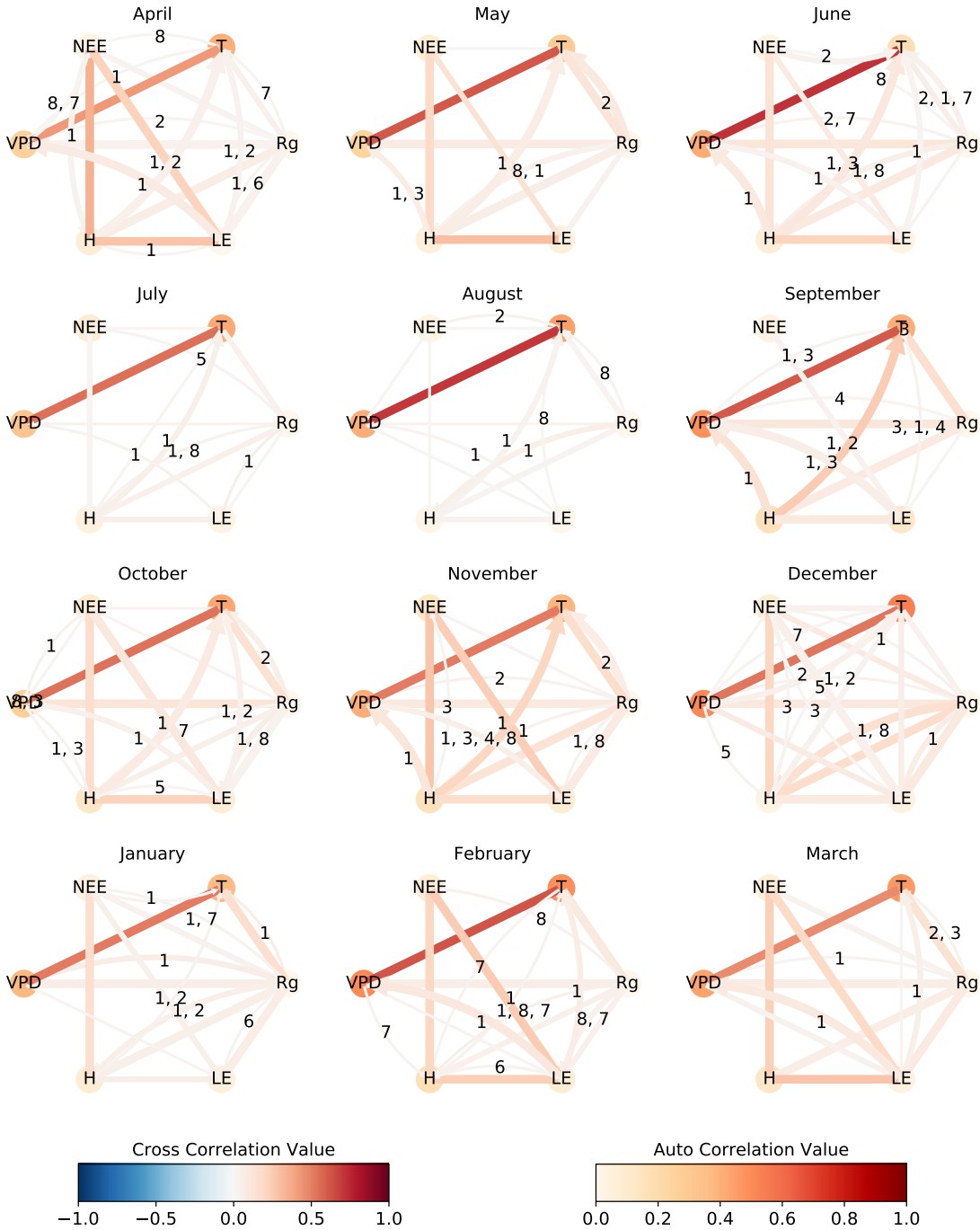

**Figure I1.** Same as Fig. 5 of the manuscript but the analysis was performed using a non-linear independence test. The number of significant occurrences of a link is given by its width. The link strength, given by the link color, is calculated by averaging the significant links of the towers. The link's lag is shown in the centre of each arrow, sorted in descending order of link strength. The resulting graphs are shown for April 2014 till March 2015. The significance threshold is 0.01. Note that GPDC only yields positive link strengths. Further, the strength values estimated with GPDC are rather weak due to the low number of datapoints and the larger sensitivity of that method to the sample size

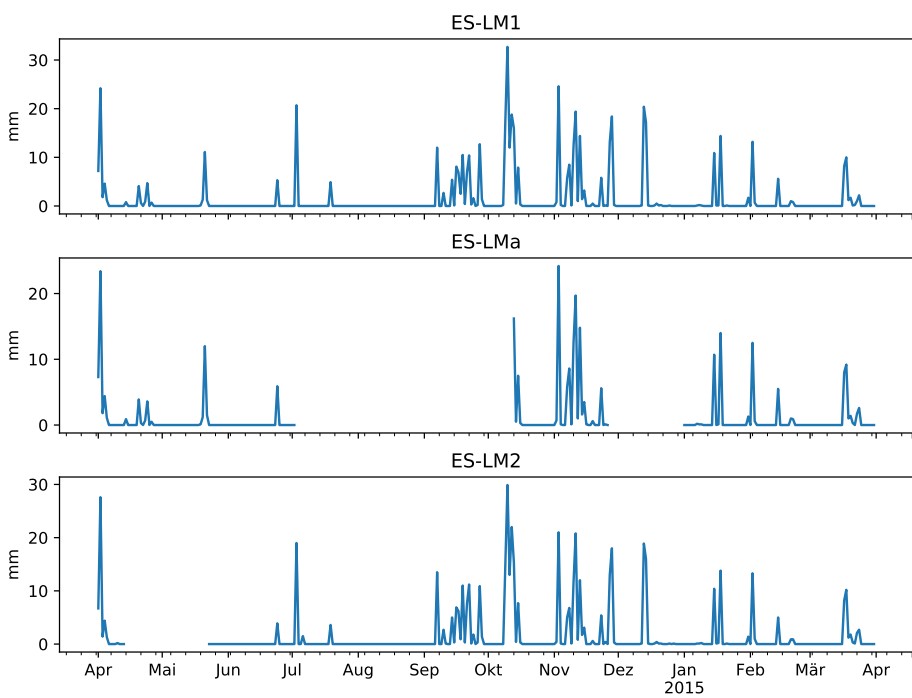

**Figure J1.** Daily aggregated precipitation in Majadas de Tiètar measured at the three tower sites from April 2014 to March 2015. Missing values are plotted as gaps.

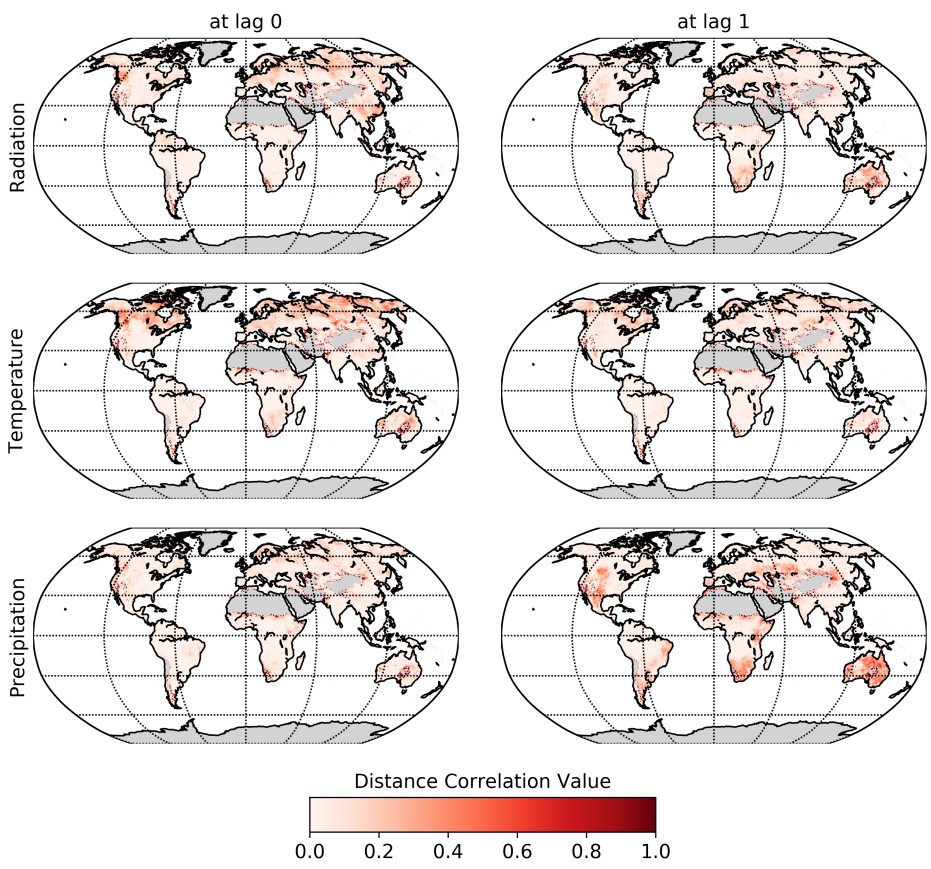

**Figure K1.** Similar to Fig. 3.3 of the manuscript. Influence of climatic drivers on NDVI as calculated by PCMCI in conjunction with the non linear independence test GPDC. The first and second columns show the estimated causal influences of climatic drivers on NDVI at lag 0 and 1, respectively.

*Author contributions.* CK and MDM designed the study with contributions from JR and DGM. CK conducted the analysis and wrote the manuscript. All authors helped to impove the manuscript. AC, MM, TEM, OPP conducted field experiments in Majadas, processed and provided its data as well helped with interpretation.

*Competing interests.* The authors declare that they have no competing interests.

5    *Acknowledgements.* We thank Maha Shadaydeh, Rune Christiansen, Jonas Peters, Milan Flach and Markus Reichstein for useful comments. CK thanks the Max Planck Research School for global Biogeochemical Cycles for supporting his PhD project. The authors affiliated with 1. thank the European Space Agency for funding the "Earth System Data Lab" project.

     This work used eddy covariance data acquired and shared by the FLUXNET community, including these networks: AmeriFlux, AfriFlux, AsiaFlux, CarboAfrica, CarboEuropeIP, CarboItaly, CarboMont, ChinaFlux, Fluxnet-Canada, GreenGrass, ICOS, KoFlux, LBA, NECC, 10   OzFlux-TERN, TCOS-Siberia, and USCCC. The ERA-Interim reanalysis data are provided by ECMWF and processed by LSCE. The FLUXNET eddy covariance data processing and harmonization was carried out by the European Fluxes Database Cluster, AmeriFlux Management Project, and Fluxdata project of FLUXNET, with the support of CDIAC and ICOS Ecosystem Thematic Center, and the OzFlux, ChinaFlux and AsiaFlux offices.

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
