# Peer review of "Estimating causal networks in biosphere-atmosphere interaction with the PCMCI approach"

_Biogeosciences, 2019_

## Referee Comment (RC1) · Anonymous Referee #1 · 4 Sep 2019

The manuscript presents the application of the newly developed PCMCI algorithm for the detection of causal links in geophysical data, focusing on biosphere/atmosphere interactions. Applications of the algorithm is done using flux tower eddy covariance data, for fine temporal scale analysis, and satellite derived NDVI along with climate data are used for global scale analysis at coarser scales. The topic is important and clearly within the scope of Biogeosciences. Identifying and quantifying causation in geophysical data is crucial for understanding the interplay of the processes involved and developing models. The present manuscript is intended to my understanding to be primarily a proof of concept of the applicability of the PCMCI algorithm. While the paper is well written several parts need to be better clarified.

Specific comments:

[Figure]

o I find the description of the algorithm on the paper slightly confusing (in particular for a Biogeosciences audience). I believe that the reader must refer to (Runge et al., 2018) to understand the basic principles behind the algorithm. I strongly suggest the authors to restructure and clarify this section. Simplifying the description and reporting the algorithm details as supplementary information could benefit the fluency of the manuscript.

o The synthetic test developed to quantify the skill of the algorithm, when its assumptions are not valid (e.g. seasonality, heteroscedacity) is clearly important. However, it is not currently clear how the results derived from this analysis can be generalized beyond the Hainich site. Emphasis should be given on the transferability of the magnitude of expected biases at a global scale.

o In a broader sense, a key question is why would the authors use a procedure for the identification of causal links, when the basic assumptions of the procedure are violated by the data?

o I believe the authors should better explain why the proposed algorithm in more efficient than e.g. the spectral Granger causality algorithm proposed earlier by Detto et al. (2012), which, to my understanding, is inherently non-parametric and not sensitive to periodicities.

o In Fig2 it is shown that with an increasing sample size, the fraction of falsely identified causal links increases, when the algorithm's assumptions are not valid. This can be a significant drawback as the best datasets (i.e. with long records), are more prone to errors. The authors should better discuss this.

o If I understood correctly, it is shown that for the baseline case, the algorithm cannot robustly identify the true causal non-linear links (as expected). However, the identification rate increases incorporating seasonality (which I presume also violates the stationarity assumption of the algorithm), which is counter intuitive. The authors in their discussion attribute this behaviour to the variance of the parent variable. Can the authors discuss

how this artefact can limit the range of applicability of the procedure for global scale applications (i.e. differences in regions with distinct seasonality or not)?

---

## Referee Comment (RC2) · Anonymous Referee #2 · 12 Sep 2019

The presented manuscript presents an interesting and novel approach to better understand biosphere-atmosphere interactions. The paper is clearly of interest for the scientific community and fits well in the scope of the Biogeosciences journal. While moving on from the classical correlation approach is needed and of great interest, because causality is modern topic and no so broadly used, the paper needs to do a better effort to introduce the topic in an easy way to the community in order to be published. Actually, it is difficult for me to review the results of the paper until the methods are more clearly exposed to the reader.

These are my specific comments:

Introduction:

[Figure]

*I miss a paragraph showing the limitations of the classical correlations analysis, when the failed, when causality approaches did better and why...

Methods:

*In general, as I said, the methods are hard to follow. I suggest to simplify/restructure the section to facilitate its understanding. The section 2.1.2 is probably the most confusing to me, I recommend to include a flowchart to visualise the algorithm.

Results:

Line 5, page 12, replace "stonger" by stronger.

Discussion:

*Lines 7-11: After reading the paper, I am still not convinced that using a linear independence test is the way to proceed. I think you have to demonstrate it with an example. Perhaps, you can run your artificial dataset tests using a non linear rank independence test (spearman's correlation) and compare the results. These results could be added to an appendix to better support your statements if that's the case.

---

## Referee Comment (RC3) · Benjamin L. Ruddell (Referee) · 14 Sep 2019

General Comments

This is Ben Ruddell writing; I waive anonymity for this review. When I saw this paper come across my desk it caught my attention, because I have been working on similar topics and also following the authors' work for several years. In general, I like this paper and after reading it I would like to see it published in this journal, with some changes. This general area of work needs a lot more attention because of the promise of the general approach and the urgency of getting our inference and modeling right for this kind of complex and coupled system. Thank you for this effort! After working on this kind of paper for more than a decade (and contemplating many reviews of my

own work) I've come to the opinion that we need to move past a focus on innovating methods and toward the challenge of showing how the methods can be used to produce actionable and fundamentally novel insights- or to test process theories in science. If we cannot use advanced inference techniques to learn about these systems or critique previously inaccessible scientific ideas, these methods will continue to fall on deaf ears, so to speak. So, I challenge the authors and anyone else listening to move forward aggressively with the intent to apply causal networks (Process Networks) and advanced inference techniques to interrogate scientific hypothesis and learn about systems. The current paper could do more along these lines, with added investment, by (for instance) comparing its statistical results with expectations from climate or ecological models, etc. Before beginning the review, based purely on the expectations raised by the very broad title of the paper, I already had a few questions about the paper. I will pose and then evaluate those questions before moving on to line by line comments.

1. Is the now-substantial body of literature on this topic adequately summarized and cited, giving credit where credit is due?

Papagiannopoulou et al is cited twice, but the similar paper Seddon et al. 2016 is not cited; please cite appropriately. Please review GeoInfoTheory.org, which has a nice list of publications on related topics (https://geoinfotheory.org/reference-list/). In particular, there are a few recent papers that should really be cited appropriately in your paper, because they are recent and narrowly within the scope of your literature review; these treat global land-atmosphere interactions and feedbacks using similar methods to your own. Please describe in your introduction, methods, and/or results as relevant, how does your work relate to these? Yu et al. 2019 in GCB is especially important. A list of references that would seem to be highly relevant follows.

Brunsell, N. A., and Anderson, M. C. (2011). "Characterizing the multi-scale spatial structure of land-atmosphere interactions with information theory." Biogeosciences Discussions 8.2.

Seddon, A. W., Macias‐Fauria, M., Long, P. R., Benz, D., & Willis, K. J. (2016). Sensitivity of global terrestrial ecosystems to climate variability. Nature, 531, 229–232. https://doi.org/10.1038/nature16986

Gerken et al. (in press) Robust observations of land-to-atmosphere feedbacks using the information flows of FLUXNET, NPJ Climate and Atmospheric Science

Garland, J. and E. Bradley (2018), Information Theory in Earth and Space Science, SIAM News, October 1st, 2018. Full Garland reference.

Gerken, T., Ruddell, B.L., Fuentes, J.D., Araujo, A., Brunsell, N.A., Maia, J., Manzi, A., Mercer, J., dos Santos, R.N., von Randow, C., and Stoy, P.C. (2017). Investigating the mechanisms responsible for the lack of surface energy balance closure in a central Amazonian tropical rainforest. Full Gerken reference.

Goodwell, Allison E., et al. "Dynamic process connectivity explains ecohydrologic responses to rainfall pulses and drought." Proceedings of the National Academy of Sciences (2018): 201800236.

Hlaváčková-Schindler, K.; Paluš, M.; Vejmelka, M.; Bhattacharya, J. (2007). Causality detection based on information-theoretic approaches in time series analysis. Phys. Rep. 441, 1–46.

James, R. G., Barnett, N. and Crutchfield, J. P. (2016) 'Information Flows? A Critique of Transfer Entropies', Physical Review Letters, 116(23).

Jiang, P. and Kumar, P. (2018). "Interactions of information transfer along separable causal paths," Phys. Rev. E 97, 042310.

Jiang, Peishi, and Praveen Kumar. "Information transfer from causal history in complex system dynamics." Physical Review E 99.1 (2019): 012306. Full Jiang and Kumar reference.

Knuth, Kevin H., et al. "Revealing relationships among relevant climate variables with

information theory." arXiv preprint arXiv:1311.4632 (2013).

Kumar, P. and Ruddell, B.L. (2010). Information Driven Ecohydrologic Self-Organization. Entropy 2010, 12, 2085–2096.

Ruddell, B. L., Yu, R., Kang, M. and Childers, D. L. (2015). 'Seasonally varied controls of climate and phenophase on terrestrial carbon dynamics: modeling eco-climate system state using Dynamical Process Networks', Landscape Ecology, pp. 1-16.

Ruddell, B.L., N.A. Brunsell and P. Stoy (2013). Applying information theory to quantify process uncertainty, feedback, and scale in the Earth system. EoS, 94, 56. Full Ruddell reference.

Smirnov, D.A. (2013). Spurious causalities with transfer entropy. Phys. Rev. E 87.

Yu, R., Ruddell, B. L., Kang, M., Kim, J., & Childers, D. (2019). Anticipating global terrestrial ecosystem state change using FLUXNET. Global change biology. https://doi.org/10.1111/gcb.14602 Full Yu reference.

2. Is the concept and any methods used for "causality" adequately posed and defended?

You can expect such a strong claim and wording as "causal graph" to be aggressively challenged by readers and reviewers in this paper and any others using the term, for a long time to come. Renaming "correlation" or "information flow" to "causation" is a major and very aggressive departure from our disciplines' wording and conceptualization during the long and mature history of statistical inference, and requires very strong justification. Granger causality has never really been "causality"; it's a type of conditional time-lagged cross correlation. Please understand my point here; I'm not asking for you to give up on the use of "causal" language, but I am strongly requesting that you spend at least a paragraph in the introduction or methods section of this paper (and others, for the foreseeable future) to argue and explain to the reader exactly what is, and is not, meant by "causal" in this context. It is otherwise too strong a term to be using. As a more general comment, it's extremely important for us to reach a consensus about what to call things. This is an iterative community process of communication that works through conversation and engagement, and through clarification about what is the same and what is different. It's not my place to decide whether we should be calling something a "causal network" or a "Process Network", but I do insist that we have the conversation. For the purposes of this paper, this means citing my recent & prior work, and that of others, and trying to explain exactly how your terms relate to our terms for similar things, and proposing what you understand the similarities and differences to be; this is particularly important when writing a methods paper such as this one under review here.

3. Is there anything new here, and is that made clear?

Yes! PCMCI is put through some rigorious tests for both satellite and 30m flux data and appears to hold up well; this is novel and interesting as a methodological development. However, in my opinion, it is important when describing this method in the methods section that you distinguish it precisely and detail from other similar methods, explaining its relative advantages and disadvantages. There are lots of other methods out there that have used Granger-adjacent directed coupling statistics in various application contexts. In this precise context, my 2009 papers you cited (and several since) used 30 minute flux tower timeseries data to determine atmo-bio networks, identifying ranges of statistically significant time lagged couplings, and also studies periodicity and noise in the method, calling these resulting patterns "Process Networks", and distinguishing the most "causally" relevant couplings using a Tz metric that compares directed vs correlative information flows. This is a well worn topic in 2019, so it's not sufficient in a methods paper to contrast your method with correlations anymore. Contrast your method precisely against others that claim similar goals and results, please. Why should we use PCMCI instead of one of several other existing similar methods? How would the results differ in theory and in practice? Should we use PCMCI in this case, and use Ruddell et al. 2009a "Tz" in another case? What are the pros and cons?

Because this paper focuses on methods, it needs to be much more specific about how these methods relate to other adjacent methods and conflicting/overlapping terminologies already in use; this engagement is how we will build our community's knowledge and practice. (your treatment of the underlying assumptions is a strength of the paper and should help make these distinctions clear; thank you for this attention to detail here.)

4. Is the very broad title justified, or is the paper actually about something much more narrow and specific?

By the end of the abstract, I decided "negative" on #4 because this paper appears to be not a review or synthesis of the broad topic of causal networks in the bio-atmo-geo-sphere as implied by the title, but instead a methods case study establishing the robustness of a proposed method MCMCI in two land-atmosphere contexts. I suggest a much narrower title, like "PCMCI robustly identifies biosphere-atmosphere interdependencies", or some such. It is very important to use an accurate title that is not over-broad. The title directly summarizes the question and/or findings, in a nutshell. An overbroad or inaccurate title is grounds for rejection in my view.

Line by Line Comments

Sec. 2.1 I've followed the derivations in Runge et al. (various, 2014-2018) and I don't have a problem with the methods. However, I have not seen here or in Runge et al. (various) an explicit comparison of the MCI approach with Ruddell and Kumar's (2009a) "Tz" or zero-lag ratio method for the disambiguation of "strongly causal" versus "common-source causal" indicated couplings. There appears to be a lot of shared intent and intuition here, and possibly some very similar (but differently named) mathematics and assumptions. Please explain what is similar or different.

Pg.20-10 This discussion on "causal stationarity" and limitation of study to one season or system state appears to be treated in Ruddell and Kumar 2009(b) (second half of the paper you cited) under the terms "local" and "global" stationarity. What's the

relationship here, please?

Pg.21-20 Although it isn't the focus of your paper, Kumar and Ruddell (2010, Entropy) and some of my more recent papers (Yu et al., Gerken et al.) have shown very strong changes in coupling strength across space, as well as across time. I wouldn't make the claim that "the interaction between biosphere and atmosphere is expected to change only marginally across space" in the absence of strong arguments supporting this. I've argued the opposite in several recent papers- I've argued that the Process Network characterizing these systems and their states changes dramatically between places and times, and that this represents a qualitative shift in how the systems are functioning. (note that I'm not arguing that physics changes, only that its structure and expression in a complex system changes dramatically) . . . please engage with this argument, or remove the claim.

Pg.21-25 Most of my papers have focused their analysis and presentation of results on a single "most significant" time lag (usually chosen as the first/shortest peak lag in my papers, called the "characteristic time lag" in my papers), or an average across a range of time lags (usually subdaily <18hrs) because of the extreme challenge of interpretation posed by a large number of statistically significant coupling links. Separating out every conditionally "momentary" coupling is not hard to do mechanically, but interpretation and communication is devilish. I think you're running into this problem here. Once we move past conditioning couplings on zero-lag correlations, it's not clear where to stop or how to interpret the results. I'd hope that PCMCI could add some clarity, but I'm not convinced based on this discussion that it is helping. Please comment and clarify if possible, or at least explain how what you're doing is different here from what Ruddell and Kumar 2009 did with T, I, Tz, canonical coupling types, and characteristic time lags. If possible, also engage with Goodwell and Kumar (recent) who have attempted to split out redundant, synergistic, and independent couplings in the land-atmosphere coupling context.

Pg.22-15 I am not convinced by biweekly or monthly scale correlation analysis in satellite or climate data represents causation in any real or approximate sense. There are several problems here. First, these data are modeled and abstracted several levels beyond primary observations, so patterns cannot be relied upon to strongly represent causal realities as well as in-situ flux data. Second, once we move past subdaily time lags, we are well into the scales dominated by diurnal cycles, synoptic weather cycles and by seasonal rhythms, so it is hard to distinguish signal from noise when the "noise" is an overwhelmingly energetic diurnal, seasonal, or synoptic cycle. Third, we already have strong reason to believe that the main process timescales are subdaily, due to e.g. our flux tower analyses, so we must presume that superdaily or monthly timescales indicated by the methods are merely echoes and confounding correlates of shorter timescale processes unless we can prove otherwise (e.g. through robust conditioning against shorter lags)- and that proof is not possible using coarse time resolution data. This is a basic problem with attempts to use satellite and coarse time resolution gridded data to establish "causal" relationships, and I haven't seen it adequately addressed in this paper or prior papers attempting similar. What am I missing here, please? Please explain how your method addresses these three problems. This gridded/monthly analysis may be a "bridge too far", so to speak, for this paper; it's different from and a weaker argument than your eddy covariance analysis, with several layers of practical problems weakening the conclusions.

Fig. 6,7 These results are begging for a detailed comparison with Yu and Ruddell et al., published earlier this year in Global Change Biology, which attempts a very similar analysis but uses an extrapolation of 30m flux data derived couplings to the global terrasphere rather than monthly gridded data. Please provide this comparison.
* * *

---

## Author Comment (AC2) · 16 Oct 2019

**1   Response to reviewer 2**

**The presented manuscript presents an interesting and novel approach to better understand biosphere-atmosphere interactions. The paper is clearly of interest for the scientific community and fits well in the scope of the Biogeosciences journal. While moving on from the classical correlation approach is needed and of great interest, because causality is modern topic and no so broadly used, the paper needs to do a better effort to introduce the topic in an easy way to the community in order to be published. Actually, it is difficult for me to review the results of the paper until the methods are more clearly exposed to the reader.**

We thank the reviewer for the support of the topic. The accessibility of the method has been criticised also by reviewer 1. In a revised version of the manuscript we will aim for an improved accessibility of the method. Please refer to your specific comment for further details.

**These are my specific comments:**

- **Introduction: I miss a paragraph showing the limitations of the classical correlations analysis, when the failed, when causality approaches did better and why**...

  We see the benefits the reviewer aims for by requesting such a paragraph. We do not claim correlation analysis to be wrong, if applied correctly. The issues arise, if one moves beyond certain boundaries within the interpretation of the results. A correlative analysis does not fulfill requirements for a causal interpretation. Any method which brings us closer to causal interpretability of a dependence structure increases the information content of the analysis. This is our motivation to test a causal inference method that is more sophisticated than the mere use of correlations.

  This argument will be further motivated, first by citing literature which showed an improved interpretability using causal methods rather than correlation (cf. Detto 2008, but also Runge2019 and others), and second by further highliting the differences of the estimated dependence structure using lagged correlation and PCMCI (cf. Fig. 1 or Fig. 4 and F1).

- **Methods: In general, as I said, the methods are hard to follow. I suggest to simplify/restructure the section to facilitate its understanding. The section 2.1.2 is probably the most confusing to me, I recommend to include a flowchart to visualise the algorithm.**

We will improve the accessibility of the method by adding an introductory paragraph to the method section. Here we explain how PCMCI relates to existing methods which will help the reader to gain a more intuitive understanding of aim and concept of PCMCI. A detailed description including mathematical notations will be then given in the following subsections. Further a graphical visualisation will also be considered.

- **Results: Line 5, page 12, replace "stonger" by stronger.**

  Thanks for noticing and pointing out this spelling mistake.

- **Discussion: Lines 7-11: After reading the paper, I am still not convinced that using a linear in-dependence test is the way to proceed. I think you have to demonstrate it with an example. Perhaps, you can run your artificial dataset tests using a non linear rank independence test (spearman's correlation) and compare the results. These results could be added to an appendix to better support your statements if that's the case.**

  In a revised manuscript we will include analysis using Gaussian-process regression and distance correlations. Preliminary, results suggest, that those outcomes are indeed very similar.

  For example, Fig. 1 is comparable to Fig. 5 from the manuscript. The difference is that Fig. 1 is calculated using Gaussian-process regressions and distance correlations as independence test (GPDC). The two figures show a similar seasonal behaviour and even good agreement in detected links. Note that GPDC only yields positive link strengths. Further, the strength values estimated with GPDC are rather weak due to the low number of data points and the larger sensitivity of that method to the sample size.

  An other example is given in Fig. 2. Here the figure is not one-to-one comparable with Fig. 6 of the manuscript because significances of an analysis using GPDC have been too (due to too low sample sizes) low to perform the same analysis.

[Figure]

Instead we plotted the link strengths of radiation, temperature and precipitation on NDVI at lag 0 and 1. At lag 0, GPDC detects some influence of temperature (and radiation) in boreal regions. At lag 1 precipitation influences mostly arid regions.

**Caption Fig. 1:**
Same as Fig. 5 of the manuscript but the analysis was performed using a non-linear independence test. The number of significant occurrences of a link is given by its width. The link strength, given by the link color, is calculated by averaging the significant links of the towers. The link's lag is shown in the centre of each arrow, sorted in descending order of link strength. The resulting graphs are shown for April 2014 till March 2015. The significance threshold is 0.01

**Caption Fig. 2:**
Influence of climatic drivers on NDVI as calculated by PCMCI in conjunction with the non linear independence test GPDC. The first and second columns show the estimated causal influences of climatic drivers on NDVI at lag 0 and 1, respectively.

[Figure]

**Fig. 1.**

Influence of climatic drivers on NDVI estimated via GPDC

at lag 0         at lag 1

Radiation

Temperature

Precipitation

Distance Correlation Value

0.0   0.2   0.4   0.6   0.8   1.0

**Fig. 2.**

---

## Author Response (AR1)

**1 Response to reviewer 1**

**The manuscript presents the application of the newly developed PCMCI algorithm for the detection of causal links in geophysical data, focusing on biosphere/atmosphere interactions. Applications of the algorithm is done using flux tower eddy covariance data, for fine temporal scale analysis, and satellite derived NDVI along with climate data are used for global scale analysis at coarser scales. The topic is important and clearly within the scope of Biogeosciences. Identifying and quantifying causation in geophysical data is crucial for understanding the interplay of the processes involved and developing models. The present manuscript is intended to my understanding to be primarily a proof of concept of the applicability of the PCMCI algorithm. While the paper is well written several parts need to be better clarified.**

We thank the reviewer for the support of the manuscript and further suggestions and address the points below.

**Specific comments:**

- **I find the description of the algorithm on the paper slightly confusing (in particular for a Biogeosciences audience). I believe that the reader must refer to (Runge et al.,2018) to understand the basic principles behind the algorithm. I strongly suggest the authors to restructure and clarify this section. Simplifying the description and reporting the algorithm details as supplementary information could benefit the fluency of the manuscript.**

  We improved the accessibility of the method by restructuring of the existing text and adding an introductory subsection to the method section. Here we explained how PCMCI relates to existing methods and what its key concept is. This helps the reader to gain a more intuitive understanding of aim and concept of PCMCI. More detailed description of assumptions, independence tests and the two parts building PCMCI are then given in the following subsections.

  Please refer to the sections 2.1, 2.2, 2.3 and 2.4 in the marked up manuscript for details.

- **The synthetic test developed to quantify the skill of the algorithm, when its assumptions are not valid (e.g. seasonality, heteroscedacity) is clearly important. However, it is not currently clear how the results derived from this analysis can be generalized beyond the Hainich site. Emphasis should be given on the transferability of the magnitude of expected biases at a global scale.**

  We believe this may be a misunderstanding. The synthetic test did not focus solely on the Hainich site. The impression might be given, as a few plots (time series and networks) are showing as an example the results from Hainich. We used radiation data from 72 FLUXNET sites to run the model. Therefore, the results are not bound to the conditions at Hainich.

  To clarify this we changed within the first paragraph of section '2.3.1 Artificial Dataset - Test Model' the sentence:

> Using time series of measured global radiation (Rg) we created three artificial time series that conceptually represent temperature (T), gross primary production (GPP) and ecosystem respiration (Reco)."

To:

> The artificial dataset was created using a test model which takes time series of measured global radiation (Rg) and creates three artificial time series that conceptually represent air temperature (T ), gross primary production (GPP) and ecosystem respiration (Reco).[....]

With further explanations in line 19ff on page 9:

> The model was fitted to real observational data (using radiation, temperature and land-atmosphere fluxes) of daily time resolution, measured by eddy-covariance method (Baldocchi et al., 1988; Baldocchi, 2003) from FLUXNET, by minimizing the sum of squared residuals using the gradient descent implemented in the Optim.jl package (Mogensen and Riseth, 2018). We fitted the model to 72 sites listed in Table B1 given in the Supplementary Material section.

We think that this adjustment should clarify that we are not focusing on the Hainich site only.

- **In a broader sense, a key question is why would the authors use a procedure for the identification of causal links, when the basic assumptions of the procedure are violated by the data?**

Every statistical method comes with a set of assumptions which are required to assure interpretability of the results. In our case the assumptions guarantee theoretically that the method will estimate the true causal graph. In a real world study case, those assumptions are typically not fully met. This can be already the case for linear regression (e.g., perfect normality). Our work addresses exactly the point of how well PCMCI works under violations of the assumptions, that is, how robust the method is regarding realistic violations occurring in practice. PCMCI in conjunction with ParCorr possesses several attributes which favour its use, as shown in our analysis: high detection power, interpretability, and computational efficiency. We aimed to identify whether PCMCI and ParCorr could in fact deliver reliable results using artificial and real data.

- **I believe the authors should better explain why the proposed algorithm in more efficient than e.g. the spectral Granger causality algorithm proposed earlier by Detto et al.(2012), which, to my understanding, is inherently non-parametric and not sensitive to periodicities.**

Thanks for pointing out the need for a more detailed comparison to other methods. This is basically in line with Reviewer 3 who, however, suggested the comparison to another approach. We totally agree on the importance

of such comparisons but regard numerical analyses evaluating other methods to be out of scope of this paper. Nevertheless, we agree that we could better describe the specific characteristics and benefits of PCMCI in conjunction with the independence test ParCorr.

PCMCI is computationally efficient, has proper significance testing and attribution of link strengths. As we observe, the seasonality is an obstacle for PCMCI and might be better handled with a spectral method. But, to the best of our knowledge, spectral Granger causality is also bound to causal stationarity. Thus, within an analysis spanning (possibly) several years and seasons, one would assume a constant network structure throughout time (same network structure in winter as in summer). However, this is not the case, as we could show in our analysis of the Majadas ecosystem.

- **In Fig2 it is shown that with an increasing sample size, the fraction of falsely identified causal links increases, when the algorithm's assumptions are not valid. This can be a significant drawback as the best datasets (i.e. with long records), are more prone to errors. The authors should better discuss this.**

To analyse long record time series data one has to question whether causal stationarity is fulfilled. As we have seen in the example of the Majadas dataset, strong differences in the dependence structure occur for different months of the year. Accounting for causal stationarity still does not conflict with analysing long data records with PCMCI, because by applying a proper mask one can estimate the network structure for one month or season within multi year time series (which is similar to our mask used in the Majadas dataset: taking only noon values but for consecutive days). Such a mask increases stationarity and therefore reduces an inflated false positive rate.

This is partially discussed already on page 20 line 3 to 20. To further clarify we will modify the submitted paper as follows:

We replace line 13 and 15:

> Much of the influence of heteroscedasticity is also removed when limiting the analysis to a specific time period, i.e.season, which makes the data causally stationarity (cf. Sect. 2.1). For example, the link from radiation to GPP vanishes in winter as there is mostly no active plant material left.

With:

> The increasing FPR with increasing time series length can further raise doubts regarding the analysis of long time series. For such an analysis, though, the assumption of causal stationarity should first be assessed. For example, the link from radiation to GPP vanishes in winter as there is mostly no active plant material left. To account for causal stationarity, the analysis should be limited to time series sections where the causal structure is expected to be similar. This is typically done by limiting the analysis to a specific time period (i.e. 'masking'), e.g. a

specific season, month, or time of the day. Such masking reduces additionally further influences of remaining seasonality or heteroscedasticity.

- **If I understood correctly, it is shown that for the baseline case, the algorithm cannot robustly identify the true causal non-linear links (as expected). However, the identification rate increases incorporating seasonality (which I presume also violates the stationarity assumption of the algorithm), which is counter intuitive. The authors in their discussion attribute this behaviour to the variance of the parent variable. Can the authors discuss how this artefact can limit the range of applicability of the procedure for global scale applications (i.e. differences in regions with distinct seasonality or not)?**

Seasonality constitutes a common driver in this model. In general, such common drivers increase the dependence among the variables and, hence, lead to a higher detection rate for true links (TPR) as well as a higher false positive rate (FPR) for absent links since this driver is not conditioned out properly. Therefore it is not counter intuitive that both the TPR and the FPR rate increase in the seasonality model. To reduce the effect of seasonality further, we suggest to use a mask or use deseasonalized time series.

We added following sentences to section 4.1.

> Seasonality and heteroscedasticity constitute violations of the stationarity assumption underlying the independence test ParCorr. Seasonality constitutes a common driver in this model. In general, such common drivers increase the dependence among the variables. and, hence, lead to a higher detection rate for true links (TPR) as well as a higher false positive rate (FPR) for absent links if this driver is not conditioned out properly. This additionally causes the TPR and the FPR rate to increase in the seasonality model. As shown in [**?**], including the cause of the non-stationarity as an exogenous driver in the analysis allows PCMCI to regress out its influence on the other variables. However, for ParCorr this is only valid if the dependence on the non-stationary driver is linear. Therefore, the regression on $\mathcal{R}g$ fails for $\mathcal{GPP}$ and $\mathcal{R}eco$ in the test model. With this ill-posed setting, the probability to detect false links increases with increasing time series length or when more periods are included.

**1 Response to reviewer 2**

**The presented manuscript presents an interesting and novel approach to better understand biosphere-atmosphere interactions. The paper is clearly of interest for the scientific community and fits well in the scope of the Biogeosciences journal. While moving on from the classical correlation approach is needed and of great interest, because causality is modern topic and no so broadly used, the paper needs to do a better effort to introduce the topic in an easy way to the community in order to be published. Actually, it is difficult for me to review the results of the paper until the methods are more clearly exposed to the reader.These are my specific comments:**

We thank the reviewer for the support of the topic. The accessibility of the method has been criticised also by reviewer 1. In a revised version of the manuscript we will aim for an improved accessibility of the method. Please refer to your specific comment for further details.

Introduction: **I miss a paragraph showing the limitations of the classical correlations analysis, when the failed, when causality approaches did better and why...**

We see the benefits the reviewer aims for by requesting such a paragraph. We do not claim correlation analysis to be wrong, if applied correctly. The issues arise, if one moves beyond certain boundaries within the interpretation of the results. A correlative analysis does not fulfill requirements for a causal interpretation. Any method which brings us closer to causal interpretability of a dependence structure increases the information content of the analysis. This is our motivation to test a causal inference method that is more sophisticated than the mere use of correlations.

This argument is further motivated, first by citing literature which showed an improved interpretability using causal methods rather than correlation (cf. Detto 2008, but also Runge2019 and others), and second by highlighting the differences of the estimated dependence structure using lagged correlation and PCMCI (cf. Fig. 1 or Fig. 4 and F1).

Methods: **In general, as I said, the methods are hard to follow. I suggest to simplify/restructure the section to facilitate its understanding. The section 2.1.2 is probably the most confusing to me, I recommend to include a flowchart to visualise the algorithm.**

We improved the accessibility of the method by restructuring of the existing text and adding an introductory subsection to the method section. Here we explained how PCMCI relates to existing methods and what its key concept is. This helps the reader to gain a more intuitive understanding of aim and concept of PCMCI. Detailed description of assumptions, independence tests and the two parts building PCMCI are then given in the following subsections. Further, we added a schematic now available as Figure A1.

Results: **Line 5, page 12, replace "stonger" by stronger.**

Thanks for noticing and pointing out this spelling mistake.

Discussion: **Lines 7-11: After reading the paper, I am still not convinced that using a linear in-dependence test is the way to proceed. I think you have to demonstrate it with an example. Perhaps, you can run your artificial dataset tests using a non linear rank independence test (spearman's correlation) and compare the results. These results could be added to an appendix to better support your statements if that's the case.**

We added analyses using Gaussian-process regression and distance correlations to the supplementary section. Spearman's correlation is not applicable as it is not an conditional independence test.

Within the introduction of the discussion, we refer to these results. Specifically, on page 19 line 11 we add:

> To further convince the reader we performed analyses on the observational datasets using Gaussian regression and distance correlation as an independence test. These results show similar patterns but due to the low sample sizes exhibit worse statistical significances.

For example, Fig. 1 is comparable to Fig. 5 from the manuscript. The difference is that 1 is calculated using Gaussian-process regressions and distance correlations as independence test (GPDC). The two figures show a similar seasonal behaviour and even good agreement in detected links. Note that GPDC only yields positive link strengths. Further, the strength values estimated with GPDC are rather weak due to the low number of datapoints and the larger sensitivity of that method to the sample size.

An other example is given in Fig. 2. Here the figure is not one-to-one comparable with Fig. 6 of the manuscript because significances of an analysis using GPDC have been too (due to too low sample sizes) low to perform the same analysis. Instead we plotted the link strengths of radiation, temperature and precipitation on NDVI at lag 0 and 1. At lag 0, GPDC detects some influence of temperature (and radiation) in boreal regions. At lag 1 precipitation influences mostly arid regions.

[Figure]

Figure 1: Same as Fig. 5 of the manuscript but the analysis was performed using a non-linear independence test. The number of significant occurrences of a link is given by its width. The link strength, given by the link color, is calculated by averaging the significant links of the towers. The link's lag is shown in the centre of each arrow, sorted in descending order of link strength. The resulting graphs are shown for April 2014 till March 2015. The significance threshold is 0.01

[Figure]

Figure 2: Influence of climatic drivers on NDVI as calculated by PCMCI in conjunction with the non linear independence test GPDC. The first and second columns show the estimated causal influences of climatic drivers on NDVI at lag 0 and 1, respectively.

**1 Review 3**

This is Ben Ruddell writing; I waive anonymity for this review. When I saw this paper come across my desk it caught my attention, because I have been working on similar topics and also following the authors' work for several years. In general, I like this paper and after reading it I would like to see it published in this journal, with some changes. This general area of work needs a lot more attention because of the promise of the general approach and the urgency of getting our inference and modeling right for this kind of complex and coupled system. Thank you for this effort! After working on this kind of paper for more than a decade (and contemplating many reviews of my own work) I've come to the opinion that we need to move past a focus on innovating methods and toward the challenge of showing how the methods can be used to produce actionable and fundamentally novel insights- or to test process theories in science. If we cannot use advanced inference techniques to learn about these systems or critique previously inaccessible scientific ideas, these methods will continue to fall on deaf ears, so to speak. So, I challenge the authors and anyone else listening to move forward aggressively with the intent to apply causal networks (Process Networks) and advanced inference techniques to interrogate scientific hypothesis and learn about systems. The current paper could do more along these lines, with added investment, by(for instance) comparing its statistical results with expectations from climate or ecological models, etc.

Dear Dr. Ruddell, thank you very much for your support and helpful advice. We fully agree that it is not enough to test and advocate for a new method only. However, in order to 'learn about these systems or critique previously inaccessible scientific ideas' the method of use has to be understood in its behaviour first. As PCMCI has not been tested or applied within the context of biosphere–atmosphere interactions to date, this was a necessary step to take before addressing specific scientific questions. The latter will be the aim of following studies, which will build upon the identified strengths of PCMCI. Further, testing or comparing the network structure between models requires non deterministic dependencies which, however, is typically not given.

In the following we try to respond accurately to your questions and comments and will try to integrate those as far as possible.

Before beginning the review, based purely on the expectations raised by the very broad title of the paper, I already had a few questions about the paper. I will pose and then evaluate those questions before moving on to line by line comments.

1. Is the now-substantial body of literature on this topic adequately summarized and cited, giving credit where credit is due? Papagiannopoulou et al is cited twice, but the similar paper Seddon et al. 2016 is not cited; please cite appropriately. Please review GeoInfoTheory.org, which has a nice list of publications on related topics (https://geoinfotheory.org/reference-list/). In

particular,there are a few recent papers that should really be cited appropriately in your paper,because they are recent and narrowly within the scope of your literature review; these treat global land-atmosphere interactions and feedbacks using similar methods to your own. Please describe in your introduction, methods, and/or results as relevant, how does your work relate to these? Yu et al. 2019 in GCB is especially important. A list of references that would seem to be highly relevant follows. [please refer to original comment to see reference list]

Thank you for this supportive comment. We improved the discussion of related literature. There we restructured paragraph 3 and 4 of the introduction which are now spanning from page 3 line 17 to page 4 line 13.

2. **Is the concept and any methods used for "causality" adequately posed and defended? You can expect such a strong claim and wording as "causal graph" to be aggressively challenged by readers and reviewers in this paper and any others using the term, fora long time to come. Renaming "correlation" or "information flow" to "causation" is a major and very aggressive departure from our disciplines' wording and conceptualization during the long and mature history of statistical inference, and requires very strong justification. Granger causality has never really been "causality"; it's a type of conditional time-lagged cross correlation. Please understand my point here; I'm not asking for you to give up on the use of "causal" language, but I am strongly requesting that you spend at least a paragraph in the introduction or methods section of this paper (and others, for the foreseeable future) to argue and explain to the reader exactly what is,and is not, meant by "causal" in this context. It is otherwise too strong a term to be using. As a more general comment, it's extremely important for us to reach a consensus about what to call things. This is an iterative community process of communication that works through conversation and engagement, and through clarification about what is the same and what is different. It's not my place to decide whether we should be calling something a "causal network" or a "Process Network", but I do insist that we have the conversation. For the purposes of this paper, this means citing my recent & prior work,and that of others, and trying to explain exactly how your terms relate to our terms for similar things, and proposing what you understand the similarities and differences to be; this is particularly important when writing a methods paper such as this one under review here.**

Indeed the prefix "Granger" is always used in Granger causality analysis to differentiate it from some stronger form of causality. The same applies to concepts of Information Flow or Transfer Entropy, which avoid the term causal. This stronger form has so far remained elusive, but it is the merit of seminal works by Judea Pearl (Pearl 2009) and others to put the term causal on a solid mathematical basis. This is the framework of causal

graphical models (Spirtes 2000, Pearl 2009) which lays out the assumptions under which graphs (aka networks) estimated based on conditional independencies can be interpreted causally. Not least through his popular science book (Pearl 2018), Pearl has advocated to overcome, as he states, the 'mental barrier' of using causal language— as long as it is used together with the assumptions required. And these assumptions have to be thoroughly discussed in any analysis. This is exactly the goal of our work. PCMCI belongs to the causal graphical model framework and under the assumptions listed in the paper (refer to Sect) and in the limit of infinite time series length PCMCI converges to the true causal graph, which is why we use the term causal. As we deal with finite sample length and partially unfulfilled assumptions we mention several times, that spurious links can appear (in excess to the expected FPR) and that each detected link has to be interpreted carefully.

This is now mentioned at the end of the new Paragraph 'Evolution of PCMCI from information theory' of the method section.

3. **Is there anything new here, and is that made clear? Yes! PCMCI is put through some rigorous tests for both satellite and 30m flux data and appears to hold up well; this is novel and interesting as a methodological development. However, in my opinion, it is important when describing this method in the methods section that you distinguish it precisely and detail from other similar methods,explaining its relative advantages and disadvantages. There are lots of other methods out there that have used Granger-adjacent directed coupling statistics in various application contexts. In this precise context, my 2009 papers you cited (and several since)used 30 minute flux tower time series data to determine atmo-bio networks, identifying ranges of statistically significant time lagged couplings, and also studies periodicity and noise in the method, calling these resulting patterns "Process Networks", and distinguishing the most "causally" relevant couplings using a Tz metric that compares directed vs correlative information flows. This is a well worn topic in 2019, so it's not sufficient in a methods paper to contrast your method with correlations anymore. Contrast your method precisely against others that claim similar goals and results, please. Why should we use PCMCI instead of one of several other existing similar methods?How would the results differ in theory and in practice? Should we use PCMCI in this case, and use Ruddell et al. 2009a "Tz" in another case? What are the pros and cons? Because this paper focuses on methods, it needs to be much more specific about how these methods relate to other adjacent methods and conflicting/overlapping terminologies already in use; this engagement is how we will build our community's knowledge and practice. (your treatment of the underlying assumptions is a strength of the paper and should help make these distinctions clear; thank you for this attention to detail here.)**

Thank you for the above comments. We improved the accessibility of the method by restructuring of the existing text and adding an introductory

subsection to the method section. Here we explained how PCMCI relates to existing methods and what its key concept is. This helps the reader to gain a more intuitive understanding of aim and concept of PCMCI. Detailed description of assumptions, independence tests and the two parts building PCMCI are then given in the following subsections.

However, an in-depth numerical comparison of the available methods is beyond the scope of the manuscript, and partially already done in Runge et al. 2018 and 2019. There are several causal inference methods available, with multiple additional modifications. Picking only one or two of them (e.g., Tz statistic of Ruddell et al. 2009) would unavoidably be rather arbitrary. We agree that this comparison is important but might be best tackled in a separate study, maybe even in a combined effort. Such a comparison study would help users choose the most suitable method for each specific application, rather than addressing any specific question with the method at hand, as it is common practice. See also the causality benchmark platform www.causeme.net which addresses method comparison on a growing number of benchmark datasets.

4. **Is the very broad title justified, or is the paper actually about something much more narrow and specific?By the end of the abstract, I decided "negative" on 4. because this paper appears to be not a review or synthesis of the broad topic of causal networks in the bio-atmo-geo-sphere as implied by the title, but instead a methods case study establishing the robustness of a proposed method MCMCI in two land-atmosphere contexts. I suggest a much narrower title, like "PCMCI robustly identifies biosphere-atmosphere interdependencies", or some such. It is very important to use an accurate title that is not over-broad. The title directly summarizes the question and/or findings, in a nutshell.An overbroad or inaccurate title is grounds for rejection in my view.**

Thank you for this advice. We renamed the title to "Estimation of causal networks in biosphere-atmosphere interaction: The PCMCI approach".

**Line by Line Comments**

Sec. 2.1 **I've followed the derivations in Runge et al. (various, 2014-2018) and I don't have a problem with the methods. However, I have not seen here or in Runge et al. (various) an explicit comparison of the MCI approach with Ruddell and Kumar's(2009a) "Tz" or zero-lag ratio method for the disambiguation of "strongly causal" versus "common-source causal" indicated couplings. There appears to be a lot of shared intent and intuition here, and possibly some very similar (but differently named) mathematics and assumptions. Please explain what is similar or different.**

PCMCI and Tz are two quite different approaches: PCMCI is a multivariate causal network estimation approach, Tz is bivariate and, at least in general, cannot exclude common causes or indirect links. Tz is a bivariate Transfer Entropy (TE) divided by the zero-lag mutual information

(MI). Ruddell and Kumar's(2009a) give a "coupling type" classification regarding the coupling interpretation of different values of TE and MI. However, any bivariate analysis is difficult to interpret causally since common drivers can both increase or decrease a MI or TE. For example, a significant MI and TE with TE ¿ MI, classified as "forcing dominated coupling" in Ruddell and Kumar's(2009a), can be the result of a common driver that drives X and Y in different ways. Hence, PCMCI and Tz are difficult to compare directly.

Pg.20-10 **This discussion on "causal stationarity" and limitation of study to one season or system state appears to be treated in Ruddell and Kumar 2009(b) (second half of the paper you cited) under the terms "local" and "global" stationarity. What's the relationship here, please?**

We studied the paper "Ecohydrologic process networks: 2. Analysis and characterization" by Ruddell and Kumar. We could not identify a definition of local or global stationarity. To our understanding the terms 'local' and 'global' are used in the context of choosing the bounds for the binning intervals in the estimation of the conditional probability densities. A local scheme refers to a binning interval chosen by the minimum and maximum values of the month. A global scheme refers to the binning interval that is chosen by the minimum and maximum values of the whole time series or dataset. The global scheme is chosen if a comparison between process networks is intended.

Causal stationarity means: A process $\mathcal{X}$ with graph $\mathcal{G}$ is called causally stationary over a time index $\mathcal{T}$, iff for all links $X_{t-\tau}^i \to X^j$ in the graph $X_{t-\tau}^i \not\perp X^j \mid \mathbf{X}_t^- \backslash \{X_{t-\tau}^i\}$ holds for all $t \in \mathcal{T}$. An example: The influence from radiation to temperature exists in both summer and winter, it might weaken or strengthen but as the physical mechanisms remain active, the link satisfies causal stationarity through out the year. In contrast, the influence of radiation on photosynthesis in a deciduous forest exist in summer but can not exist in winter if no photoactive plant material is present. Thus causal stationarity is violated if the whole year is included in the analysis. Limiting the analysis to specific periods in time, e.g. summer, leads to causal stationarity. This masking in time in PCMCI could be done manually/fixed time intervals, e.g. monthly, or by choosing the mask for a specific value range of one specific variable, i.e. GPP or Rg. The latter might remind of the above mentioned local and global binning but still only marginally, from our perspective.

We noticed that this explanation is missing in the method section. We added it accordingly.

Pg.21-20 **Although it isn't the focus of your paper, Kumar and Ruddell (2010, Entropy) and some of my more recent papers (Yu et al., Gerken et al.) have shown very strong changes in coupling strength across space, as well as across time. I wouldn't make the claim that "the interaction between biosphere and atmosphere is expected to change only marginally across space" in the absence of strong arguments supporting this. I've argued the opposite in several recent papers- I've argued that the Process**

**Network characterizing these systems and their states changes dramatically between places and times, and that this represents a qualitative shift in how the systems are functioning. (note that I'm not arguing that physics changes, only that its structure and expression in a complex system changes dramatically)...please engage with this argument,or remove the claim.**

The claim "the interaction between biosphere and atmosphere is expected to change only marginally across space" was used only in the context of the Majadas ecosystem and within this context we regard it justified and well supported: This ecosystem is a rather homogeneous Savanna. Within this ecosystem three eddy-covariance towers are situated within a distance of up to one kilometre (app.). At such spatial scale, climatic conditions are very similar. Due to the homogeneity of the ecosystem "interaction between biosphere and atmosphere is expected to change only marginally across space".

We clarified that this statement by adding 'across space within this ecosystem' to the sentence 'Thus, also the interaction between biosphere and atmosphere is expected to change only marginally across space within this ecosystem'.

We again emphasize that PCMCI and the Tz measure used in the referees studies cannot be directly compared and any discussion on "strong" or "weak" couplings has to take into account whether the measure is multivariate or bivariate, since excluding the effect of common drivers can strongly change the value of a measure.

Pg.21-25 **Most of my papers have focused their analysis and presentation of results on a single "most significant" time lag (usually chosen as the first/shortest peak lag in mypapers, called the "characteristic time lag" in my papers), or an average across a range of time lags (usually subdaily ¡18hrs) because of the extreme challenge of interpretation posed by a large number of statistically significant coupling links. Separating out every conditionally "momentary" coupling is not hard to do mechanically, but interpretation and communication is devilish. I think you're running into this problem here. Once we move past conditioning couplings on zero-lag correlations, it's not clear where to stop or how to interpret the results. I'd hope that PCMCI could add some clarity, but I'm not convinced based on this discussion that it is helping. Please comment and clarify if possible, or at least explain how what you're doing is different here from what Ruddelland Kumar 2009 did with T, I, Tz, canonical coupling types, and characteristic timelags. If possible, also engage with Goodwell and Kumar (recent) who have attempted to split out redundant, synergistic, and independent couplings in the land-atmosphere coupling context.**

We agree that interpreting a process network incorporating many lags for one dependence can pose difficulties. That is why we omitted a detailed analysis/study of the monthly Majadas networks. Yet, we also did not want to aggregate or focus on one lag as this would have reduced the

information content of the analysis. An averaging of lagged links, for example, would have caused a strong deviation in link strength for the dependence H→VPD in Fig. 4 (August) among the towers. $I(T→VPD)_{LMa}$ would be nearly 0 while $I(T→VPD)_{LM1}$ and $I(T→VPD)_{LM2}$ would be around 0.25. This is due to the possibility of negative coupling strengths using ParCorr. The dependence H→VPD appears at lag 1 and 3. The confidence intervals of the strength values from the three towers are overlapping for both lags, but as the link H→VPD at lag 3 is rather weak, only one crosses the significance threshold.

Defining the maximum lag might be indeed difficult from a physiological/physical perspective. In Runge2018a following is suggested: "Choice of $\tau_{max}$: The maximum time delay depends on the application and should be chosen according to the maximum physical time lag expected in the complex system. In practice we recommend a rather large choice that includes peaks in the lagged cross-correlation function (or a more general measure corresponding to the chosen independence test), because a too large choice of $\tau_{max}$ merely leads to longer runtimes of PCMCI, but not to an increased estimation dimension as for FullCI."

Due to the differences between PCMCI and the Tz statistic laid out above, we omit further comparison.

Pg.22-15 **I am not convinced by biweekly or monthly scale correlation analysis in satellite or climate data represents causation in any real or approximate sense. There are several problems here. First, these data are modeled and abstracted several levels beyond primary observations, so patterns cannot be relied upon to strongly represent causal realities as well as in-situ flux data. Second, once we move past subdaily timelags, we are well into the scales dominated by diurnal cycles, synoptic weather cy-cles and by seasonal rhythms, so it is hard to distinguish signal from noise when the "noise" is an overwhelmingly energetic diurnal, seasonal, or synoptic cycle. Third, we already have strong reason to believe that the main process timescales are subdaily, due to e.g. our flux tower analyses, so we must presume that super daily or monthly timescales indicated by the methods are merely echoes and confounding correlates of shorter timescale processes unless we can prove otherwise (e.g. through robust conditioning against shorter lags)- and that proof is not possible using coarse time resolution data. This is a basic problem with attempts to use satellite and coarse time resolution gridded data to establish "causal" relationships, and I haven't seen it adequately addressed in this paper or prior papers attempting similar. What am I missing here, please? Please explain how your method addresses these three problems. This gridded/monthly analysis may be a "bridge too far", so to speak, for this paper; it's different from and a weaker argument than your eddy covariance analysis, with several layers of practical problems weakening the conclusions.**

This comment might be addressed by addressing comment 2. Causal relationships are best examined by perturbing the system at a specific time and state (variable) (do calculus of Pearl). Though such experiments are usually not feasible in a controlled manner within Earth system sciences. Therefore, we (as a community) rely on the estimation of causal dependencies from time series and can only detect the signal imprinted in the time series. The signal of interactions depends on properties of the interaction itself, i.e. the strength,type and pattern, but also the signal-to-noise ratio, i.e. measurement noise, time sampling intervals and time aggregation. Therefore, the signal of interactions detectable within the time series (i.e. dependence within the conditional probability distributions, which using Markov condition determines connectedness in graph) might not correspond to the actual physical interactions anymore, but might very well allow valuable insight. Especially when trying to evaluate and compare dependence structures within model time series. Further, under aggregation information from fast interactions will be lost (and maybe visible as contemporaneous interactions in our networks) but processes which are dominant on larger time scales might appear as their signal is improved due to aggregation.

Further, Ruddell and Kumar 2009 and Krich 2019 find links on timescales below 30 min on 30 min time resolution fluxdata. Having the above in mind, i.e. keeping in mind that the time sampling interval determines the appearance of the causal graph, one can not even speak of the 'true causal relationships' using 30min resolution data. If links appear that happen on faster time scales than the time resolution, they will be shown as contemporaneous links (undirected) in our networks. In the method section, we state, that spurious links, both contemporaneous and lagged, can appear. This will be further elaborated in a revised manuscript.

Fig. 6,7 **These results are begging for a detailed comparison with Yu and Ruddell etal., published earlier this year in Global Change Biology, which attempts a very similar analysis but uses an extrapolation of 30m flux data derived couplings to the global terrasphere rather than monthly gridded data. Please provide this comparison.**

With all respect, we do not fully agree on the level of similarity between these two studies. Without a doubt the performed study "Anticipating global terrestrial ecosystem state change using FLUXNET" by Yu and Ruddell et al. 2019 is very interesting and we actually had similar ideas for another study. To explain why we prefer to omit a comparison with this study, we briefly summarize the method and subsequently give the reasoning.

Yu and Ruddell (2019) calculated two bivariate transfer entropy couplings (Temp-NEE, Precip-NEE) on monthly time periods of the available time series data of 204 Fluxnet towers. Thus they obtain a network per month which can be translated to monthly time series of couplings. These couplings are fitted with a specific model (using monthly averages or sums of Rg, Temp, Precip, EVI) to estimate an elasticity of that coupling to each 'driver'. Those elasticities are upscaled to global scale using an artificial neural network. Those maps of upscaled elasticities shall be compared to

PCMCI strength values.

The choice of variables for the coupling calculations are based upon: (quote from the paper) "an eddy covariance tower's process network can be approximated using three functional subsystems: Synoptic, Atmospheric boundary layer (ABL), and Turbulent. We choose an essential timeseries from each of those three subsystems: for the Synoptic subsystem, air temperature; for the ABL subsystem, precipitation; and for the Turbulent subsystem, net ecosystem exchange of carbon".

We believe a comparison is not straightforward because of two reasons: The quantity we plot in Fig. 6 and 7 is a conditional independence measure, i.e. partial correlation coefficient between time series residuals at monthly resolution in a multivariate analysis. The plotted elasticities in Fig. 3 of Yu and Ruddell et al. 2019 represent an upscale of a specific co variation (an exponential model) of a conditional independence measure, i.e. bivariate transfer entropy between time series at 30 min resolution, to monthly aggregates of climate and phenology variables. We have difficulties to relate these two quantities with each other. Furthermore, We want to validate the outcome of PCMCI. Thus we preferably compare our results to studies which calculate a dependence measure on approximately the same data as we used.

Second, Fig. 6 and 7 of our study show the dependence of phenology (NDVI) on climatic drivers. Fig. 3 of Yu and Ruddell et al. 2019 shows the elasticities of NEE to both climatic and phenological drivers. Fluctuations and responses of NEE and NDVI to climatic factors happen on very different time scales. Further, NEE and NDVI are difficult to compare in the first place.

We hope that we could convince you that the comparison of our global case study to Wu et al. (2015) and Papagiannopoulou et al. (2017b) is better suited for verification purposes than a comparison to Yu and Ruddell et al. 2019.

[revised manuscript text omitted]

Imer, D., Merbold, L., Eugster, W., and Buchmann, N.: Temporal and Spatial Variations of Soil $CO_2$, $CH_4$ and $N_2O$ Fluxes at Three Differently

5    Managed Grasslands, Biogeosciences, 10, 5931–5945, https://doi.org/https://doi.org/10.5194/bg-10-5931-2013, 2013.

Jacobs, C. M. J., Jacobs, A. F. G., Bosveld, F. C., Hendriks, D. M. D., Hensen, A., Kroon, P. S., Moors, E. J., Nol, L., Schrier-Uijl, A., and Veenendaal, E. M.: Variability of Annual $CO_2$ Exchange from Dutch Grasslands, Biogeosciences, 4, 803–816, https://doi.org/https://doi.org/10.5194/bg-4-803-2007, 2007.

James, R. G., Barnett, N., and Crutchfield, J. P.: Information Flows? A Critique of Transfer Entropies, 
[revised manuscript text omitted]

Runge, J., Heitzig, J., Petoukhov, V., and Kurths, J.: Escaping the Curse of Dimensionality in Estimating Multivariate Transfer Entropy, Phys. Rev. Lett., 108, 258 701, https://doi.org/10.1103/PhysRevLett.108.258701, https://link.aps.org/doi/10.1103/PhysRevLett.108.258701, 2012b.

Runge, J., Petoukhov, V., and Kurths, J.: Quantifying the Strength and Delay of Climatic Interactions: The Ambiguities of Cross Correlation and a Novel Measure Based on Graphical Models, Journal of Climate, 27, 720–739, https://doi.org/10.1175/JCLI-D-13-00159.1, https://doi.org/10.1175/JCLI-D-13-00159.1, 2014.

Runge, J., Nowack, P., Kretschmer, M., Flaxman, S., and Sejdinovic, D.: Detecting causal associations in large nonlinear time series datasets, arXiv e-prints, arXiv:1702.07007v2, https://arxiv.org/abs/1702.07007v2, 2018.

Runge, J., Bathiany, S., Bollt, E., Camps-Valls, G., Coumou, D., Deyle, E., Glymour, C., Kretschmer, M., Mahecha, M. D., Muñoz-Marí, J., et al.: Inferring causation from time series in Earth system sciences, Nature communications, 10, 2553, 2019.

Saleska, S. R., Miller, S. D., Matross, D. M., Goulden, M. L., Wofsy, S. C., da Rocha, H. R., de Camargo, P. B., Crill, P., Daube, B. C., de Freitas, H. C., Hutyra, L., Keller, M., Kirchhoff, V., Menton, M., Munger, J. W., Pyle, E. H., Rice, A. H., and Silva, H.: Carbon in Amazon Forests: Unexpected Seasonal Fluxes and Disturbance-Induced Losses, Science, 302, 1554–1557, https://doi.org/10.1126/science.1091165, 2003.

Schade, G. W., Goldstein, A. H., and Lamanna, M. S.: Are Monoterpene Emissions Influenced by Humidity?, Geophysical Research Letters, 26, 2187–2190, https://doi.org/10.1029/1999GL900444.

Schreiber, T.: Measuring Information Transfer, Phys. Rev. Lett., 85, 461–464, https://doi.org/10.1103/PhysRevLett.85.461, https://link.aps.org/doi/10.1103/PhysRevLett.85.461, 2000.

Scott, R. L., Huxman, T. E., Cable, W. L., and Emmerich, W. E.: Partitioning of Evapotranspiration and Its Relation to Carbon Dioxide Exchange in a Chihuahuan Desert Shrubland, Hydrological Processes, 20, 3227–3243, https://doi.org/10.1002/hyp.6329, 2006.

Scott, R. L., Cable, W. L., and Hultine, K. R.: The Ecohydrologic Significance of Hydraulic Redistribution in a Semiarid Savanna, Water Resources Research, 44, https://doi.org/10.1029/2007WR006149, 2008.

Shadaydeh, M., Garcia, Y. G., Mahecha, M., Reichstein, M., and Denzler, J.: Analyzing the Time Variant Causality in Ecological Time Series: A Time-Frequency Approach, in: International Conference on Ecological Informatics (ICEI), pp. 151–152, 2018.

Shannon, C.: A Mathematical Theory of Communication, Bell System Technical Journal, 27, 379–423, https://doi.org/10.1002/j.1538-7305.1948.tb01338.x, https://www2.scopus.com/inward/record.uri?eid=2-s2.0-84940644968&doi=10.1002%2fj.1538-7305.1948.tb01338.x&partnerID=40&md5=8164c9f2519a48647f6ea2d501b98177, cited By 13847, 1948.

Smirnov, D. A.: Spurious causalities with transfer entropy, Phys. Rev. E, 87, 042 917, https://doi.org/10.1103/PhysRevE.87.042917, https://link.aps.org/doi/10.1103/PhysRevE.87.042917, 2013.

[revised manuscript text omitted]